# CPiRi: Channel Permutation-Invariant Relational Interaction for Multivariate Time Series Forecasting

**Jiyuan Xu, Wenyu Zhang, Xin Jing, Jiahao Nie, Shuai Chen, Shuai Zhang**[*]
School of Information Technology and Artificial Intelligence
Zhejiang University of Finance and Economics
Hangzhou, China
`{straka1, jingxin, mozo, zhangshuai}@zufe.edu.cn`
`wyzhang@e.ntu.edu.sg, jhnie@hdu.edu.cn`

## Abstract

Current methods for multivariate time series forecasting can be classified into channel-dependent and channel-independent models. Channel-dependent models learn cross-channel features but often overfit the channel ordering, which hampers adaptation when channels are added or reordered. Channel-independent models treat each channel in isolation to increase flexibility, yet this neglects inter-channel dependencies and limits performance. To address these limitations, we propose **CPiRi**, a **channel permutation invariant (CPI)** framework that infers cross-channel structure from data rather than memorizing a fixed ordering, enabling deployment in settings with structural and distributional co-drift without retraining. CPiRi couples **spatio-temporal decoupling architecture** with **permutation-invariant regularization training strategy**: a frozen pretrained temporal encoder extracts high-quality temporal features, a lightweight spatial module learns content-driven inter-channel relations, while a channel shuffling strategy enforces CPI during training. We further **ground CPiRi in theory** by analyzing permutation equivariance in multivariate time series forecasting. Experiments on multiple benchmarks show state-of-the-art results. CPiRi remains stable when channel orders are shuffled and exhibits strong **inductive generalization** to unseen channels even when trained on **only half** of the channels, while maintaining **practical efficiency** on large-scale datasets. The source code is released at JasonStraka/CPiRi.

## 1 Introduction

Multivariate time series forecasting (MTSF) is critical in domains like finance and transportation, where modeling inter-channel relationships is essential (Zhang et al., 2025). Research in this area has largely bifurcated into two paradigms with a paradoxical trade-off: channel-independent (CI) and channel-dependent (CD) models (Shao et al., 2025b).

CD models, spanning architectures from graph neural networks (GNNs) to Transformers, explicitly model relational interactions across channels. Despite their sophisticated designs, many exhibit a critical limitation: they overfit to the static positional configurations of training data rather than learning semantic relationships. As illustrated in Fig. 1, these models memorize channel order instead of content-driven dependencies. Consequently, they suffer catastrophic degradation when encountering channel permutations or new channels during inference, which are common scenarios in real-world dynamic systems and typical of structural co-drift in real deployments (e.g., sensor networks or evolving financial metrics). This positional rigidity fundamentally undermines their robustness and adaptability. Conversely, CI models (e.g., DLinear (Zeng et al., 2023), PatchTST (Nie et al., 2023)) process channels independently, ensuring robustness against noise and *channel heterogeneity* (Shao et al., 2025b). However, they ignore cross-channel interactions, sacrificing the core advantage of multivariate analysis and limiting forecasting performance.

---

[*]Corresponding author

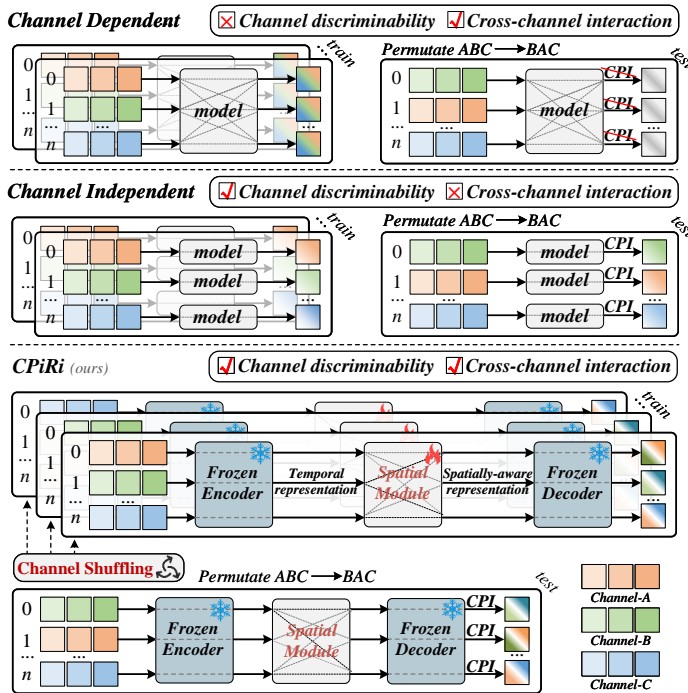

Figure 1: Conceptual overview of the CPiRi framework. Top: channel-dependent models often overfit to channel order, leading to performance collapse in channel permutation invariance (CPI) tests. Middle: channel-independent models lack cross-channel interaction. Bottom: CPiRi resolves this by (1) using a frozen encoder for robust temporal feature extraction, and (2) training a channel permutation-equivariant spatial module with a channel shuffling strategy to learn generalizable, content-based relationships, ensuring both high accuracy and near-invariant robustness under channel shuffling.

This dichotomy presents a fundamental dilemma: CD models capture essential interactions but lack adaptability, while CI models ensure robustness at the cost of relational reasoning. To address this, we posit that models should achieve channel permutation invariance (CPI), i.e. maintaining performance regardless of channel ordering or additions, without compromising interaction modeling. To validate this need, we introduce a CPI diagnostic: models genuinely understanding channel relationships should exhibit stability under channel shuffling. Our tests reveal alarming fragility in state-of-the-art (SOTA) CD models. For instance, on PEMS-08 (Sec. 4.2), Informer's error increases by >400% under channel shuffling, confirming reliance on positional memorization rather than relational reasoning. The ultimate goal of MTSF is not merely to pass a permutation test but to develop models that are both accurate and generalizable. The CPI diagnostic serves as a crucial litmus test that exposes index- and topology-dependence, a failure mode that blocks deployment under co-drift.

In this paper, we propose CPiRi (**C**hannel **P**ermutation-**I**nvariant **R**elational **I**nteraction), a novel framework that models cross-channel dependencies in a CPI manner while synergizing CI and CD strengths. CPiRi integrates two core innovations: spatio-temporal decoupling based model architecture and permutation-invariant regularization based training strategy. The architecture consists of a frozen pre-trained temporal encoder and a lightweight, trainable spatial module. The encoder extracts channel-specific features independently, preserving CI advantages (robustness, noise immunity). The spatial module then models relational interactions across channels using only content-based cues, eliminating positional bias. This decoupling isolates temporal learning from relational reasoning, enabling the model to inherit CI stability while capturing CD interactions. The permutation-invariant regularization based training strategy employs dynamic channel shuffling during training, which removes positional shortcuts and forces the spatial module to learn content-driven relational reasoning. By exposing the model to diverse channel configurations, it acquires a meta-skill for generalizable interaction modeling, ensuring CPI without sacrificing accuracy.

Our contributions are as follows:

- The CPiRi framework, resolving the CI-CD trade-off via channel permutation-invariant relational interaction.
- A spatio-temporal decoupled architecture combining a frozen temporal encoder (CI) and content-aware spatial module (CD).
- A permutation-invariant regularization strategy using channel shuffling to enforce content-driven relational reasoning.
- SOTA results on benchmarks with: negligible degradation under channel shuffling; strong inductive generalization to unseen channels (e.g., trained on only half the channels) without retraining; and improved robustness in low-data regimes, where the regularization strategy is most beneficial.

## 2 RELATED WORK

**Channel-dependent models.** CD approaches represent the mainstream of MTSF research. However, early methods like GNNs (e.g., MTGNN (Wu et al., 2020)) learn rigid, static graphs, while subsequent Transformer-based models (e.g., Crossformer (Zhang & Yan, 2023)) often overfit to channel order by treating channels as a fixed sequence. While many recent studies propose sophisticated architectures to improve robustness or generalization, such as through adaptive hypergraphs (Chen et al., 2025), channel-aligned transformers (Xue et al., 2024), dual unmixing structures (Zhu et al., 2024), context-aware transformers (Hong et al., 2025), or by focusing explicitly on latent space generalization (Deng et al., 2024), they tend to add model complexity without resolving the fundamental positional dependency. Furthermore, recent large models like Timer-XL (Liu et al., 2024b) are ill-suited for typical MTSF tasks. They demand extensive pre-training and are computationally expensive, leading to poor performance on smaller datasets and practical scaling issues due to memory constraints. The framework of CPiRi, while primarily designed for CPI, offers a critical side-benefit: it resolves this dilemma by avoiding the need for massive data and complex spatial pre-training, making it a more practical and effective solution.

**Channel-independent models.** CI approaches, exemplified by DLinear (Zeng et al., 2023) and the SOTA PatchTST (Nie et al., 2023), prioritize robustly modeling each channel's temporal patterns independently. While efficient and inherently permutation-invariant, their fundamental limitation is the neglect of cross-channel dynamics. A new frontier involves large foundation models like Chronos-Bolt (Ansari et al., 2024) and Sundial (Liu et al., 2025). These models are often discussed in the broader context of applying large model concepts to time series analysis (Jin et al., 2024) and are trained on diverse datasets (Shao et al., 2025a), with some proposing novel objectives like predicting curve shapes (Feng et al., 2024). However, our work is one of the first to effectively integrate a powerful, pre-trained univariate foundation model (Sundial) into a multivariate task. By using Sundial as a frozen feature extractor, CPiRi isolates the challenge of relational learning, allowing a smaller, specialized module to focus entirely on learning generalizable spatial interactions.

## 3 METHODOLOGY

### 3.1 PROBLEM DEFINITION

The standard MTSF problem can be formalized as follows. For a multivariate time series containing $C$ channels (also referred to as variables or nodes), given historical data $\mathcal{X} = \{X_1, ..., X_L\} \in \mathbb{R}^{L \times C}$, where $X_t \in \mathbb{R}^C$ is the vector of values for all channels at time step $t$, and $L$ is the look-back window size. The time series forecasting task is to predict the values $\mathcal{Y} = \{X_{L+1}, ..., X_{L+T}\} \in \mathbb{R}^{T \times C}$ for the next $T$ time steps.

### 3.2 FRAMEWORK OVERVIEW

As illustrated in Fig. 2, the CPiRi framework processes multivariate time series through three distinct, sequential stages, strategically isolating temporal and spatial learning. This decoupled design allows us to leverage powerful, pre-trained temporal priors while focusing the training entirely on learning a generalizable relational reasoning skill. Then the permutation-invariant regularization strategy forces the model to learn generalizable, content-based relationships, moving beyond memorizing channel order.

Figure 2: Architectural overview of the CPiRi framework. CPiRi operates in three stages: (1) A frozen univariate foundation model (Sundial Encoder) independently extracts temporal features for each channel. (2) A lightweight, trainable spatial module processes the set of channel representations to model permutation-invariant interactions. (3) The frozen Sundial Decoder independently generates the final forecast for each channel from its updated representation.

## 3.3 RADICAL SPATIO-TEMPORAL DECOUPLING MODEL

The three-stage architecture embodies our philosophy of decoupling, strategically leveraging a pretrained model for temporal features and a lightweight module for spatial relationships.

**Stage 1: universal temporal feature extraction.** For a given input $\mathcal{X} \in \mathbb{R}^{L \times C}$, each of the $C$ channels is processed in isolation by the frozen encoder of the Sundial foundation model (Liu et al., 2025). From the encoder's output, we extract the final patch representation for each channel, yielding a set of temporal feature vectors $\{\mathbf{h}_1, \ldots, \mathbf{h}_C\}$, where each $\mathbf{h}_i \in \mathbb{R}^D$ is a $D$-dimensional feature vector.

**Stage 2: permutation-equivariant spatial interaction.** This stage is the core trainable component of CPiRi. The set of temporal representations $\{\mathbf{h}_1, \ldots, \mathbf{h}_C\}$ is treated as an unordered set and fed into a lightweight spatial module, which consists of a standard Transformer encoder block. The multi-head self-attention mechanism within this block is inherently permutation-equivariant, making the architecture structurally robust to the order of its inputs. The sole purpose of this module is to learn a generalizable function that models inter-channel dynamics based on the content of the feature vectors. The output is a set of spatially-aware representations $\{\mathbf{h}'_1, \ldots, \mathbf{h}'_C\}$, where each vector has been enriched with context from other relevant channels.

**Stage 3: independent prediction generation.** In the final stage, each enriched representation $\mathbf{h}'_i$ is independently passed to the frozen Sundial (Liu et al., 2025) decoder to generate the forecast result. This final independent step reinforces the model's robustness by preventing structural entanglement at the generation stage, thereby completing our decoupled design.

This decoupling and frozen design overcomes two fundamental limitations. First, it transfers robust temporal priors learned from large-scale external datasets, effectively mitigating data scarcity in dynamic MTSF tasks. Second, it establishes a modular paradigm where the spatial module focuses exclusively on learning cross-channel dynamics. This specialization avoids the prohibitive computational cost of retraining monolithic architectures while simultaneously addressing inefficiency in capturing sparse relational patterns.

## 3.4 PERMUTATION-INVARIANT REGULARIZATION STRATEGY

The key to unlocking CPiRi's generalization capability is its training strategy, which we frame as a form of meta-learning. By exposing the model to a distribution of different channel orderings, we compel it to learn a transferable "meta-skill" for relational reasoning that is agnostic to the arbitrary indexing of channels. The procedure is detailed in Algorithm 1. During each training step, we apply a random permutation $\pi$ to the channels of both the input batch $X$ and the target batch $Y$. The frozen temporal encoder is unaffected, as it processes channels independently. The spatial module, however, receives a randomly ordered set of channel representations. To consistently minimize the loss, the module cannot rely on positional cues (e.g., "the 3rd channel is always noisy"). Its only recourse is to learn a function that identifies relationships based on the intrinsic content of the temporal feature vectors themselves. This forces the model to transition from memorizing static correlations to learning a generalizable relational reasoning capability. While our training strategy shares the high-level goal of improving generalization (similar to meta-learning), it operates through invariance-focused data augmentation rather than task adaptation.

---

**Algorithm 1** Permutation-invariant regularization strategy via channel shuffling

---

**Require:** Training dataset $\mathcal{D}_{\text{train}}$ with $C$ channels
1: **for** each batch $(X, Y) \in \mathcal{D}_{\text{train}}$ **do**
2:     Generate a random permutation $\pi \leftarrow \Pi_C$
3:     Apply the **same** permutation: $X_\pi, Y_\pi$
4:     Compute prediction: $\hat{Y}_{\text{pred}} \leftarrow \text{CPiRi}(X_\pi)$
5:     Compute loss: $\mathcal{L} \leftarrow \text{Loss}(\hat{Y}_{\text{pred}}, Y_\pi)$
6:     Back-propagate $\mathcal{L}$ to update spatial module
7: **end for**

---

## 3.5 THEORETICAL FOUNDATION: GENERALIZABLE REASONING VIA PERMUTATION INVARIANCE

The empirical success of channel shuffling is grounded in the mathematical principle of permutation invariance. This section provides the theoretical argument for how our methodology guarantees that CPiRi learns generalizable, content-based relationships.

**The principle of permutation-equivariant.** Let $\mathcal{H} = \{\mathbf{h}_1, \ldots, \mathbf{h}_C\}$ be the set of temporal features from Stage 1. The spatial module is a function $f : (\mathbb{R}^D)^C \to (\mathbb{R}^D)^C$ that maps $\mathcal{H}$ to contextualized representations $\mathcal{H}'$. A function $f$ is *permutation-equivariant* if for any permutation $\pi$, it holds that $f(\mathbf{h}_{\pi(1)}, \ldots, \mathbf{h}_{\pi(C)}) = (f(\mathcal{H})_{\pi(1)}, \ldots, f(\mathcal{H})_{\pi(C)})$. In essence, permuting the inputs only permutes the outputs in the same way, as illustrated in Fig. 3. Without explicit constraints, a standard CD model might learn a non-equivariant function by overfitting to positional artifacts (e.g., learning a specific mapping for the 3rd channel), leading to catastrophic failure in our channel shuffling test. Since the encoder/decoder of CPiRi act independently on channels (invariant) and the spatial module is equivariant, the full pipeline of CPiRi is equivariant by closure.

**Enforcing CPI through permutation-invariant regularization.** Our training objective is to minimize the expected loss over the distribution of all possible permutations $\Pi_C$:

$$\min_\theta \mathbb{E}_{(\mathcal{X}, \mathcal{Y}) \sim \mathcal{D}, \ \pi \sim \Pi_C} \left[ \mathcal{L}(f_\theta(\mathcal{X}_\pi), \mathcal{Y}_\pi) \right]$$

where $\theta$ represents the trainable parameters of the spatial module $f$. Any non-equivariant component within $f_\theta$ that relies on a specific ordering will incur high loss for most permutations, failing to minimize the expected loss. The only stable solution is one that is inherently equivariant. Thus, the optimization process naturally drives the learned function $f_\theta$ to be permutation-equivariant.

**From invariance to relational learning.** Foundational work like Deep Sets (Zaheer et al., 2017) has shown that any permutation-equivariant function on a set must be decomposable into the form $f(\mathbf{h}_i) = \rho(\mathbf{h}_i, \bigoplus_{j=1}^{C} \phi(\mathbf{h}_j))$, where $\phi$ is an element-wise transformation, $\bigoplus$ is a symmetric aggregation function (e.g., sum, mean), and $\rho$ is a combination function. The self-attention mechanism is a canonical implementation of this structure, as it computes the output for an element $\mathbf{h}_i$ as a weighted sum of transformations of all elements in the set, with weights determined by content-based similarity.

In conclusion, a direct logical chain links our methodology to its outcome: the channel shuffling strategy (the algorithm) necessitates a permutation-equivariant function (the mathematical property), which in turn requires an architecture based on symmetric aggregation (the structural form), for which self-attention is the ideal implementation. This guarantees, by design, that CPiRi is forced to learn inter-channel relationships based on semantic content, instead of positional artifacts.



Figure 3: Channel-level permutation equivariance. Rotating or reflecting the same directed graph permutes node indices while preserving edges, yielding an equivalent adjacency matrix. This illustrates that MTSF is permutation-equivariant at the channel level, and that CPI models can adapt to dynamic multivariate time series settings with reordered, added, or removed channels.

# 4 EXPERIMENTS

## 4.1 EXPERIMENTAL SETUP

**Datasets.** Our evaluation is conducted on five widely-used public benchmark datasets: METR-LA (Li et al., 2018), PEMS-BAY (Li et al., 2023), PEMS-04, PEMS-08 (Chen et al., 2001), SD (a subset of LargeST (Liu et al., 2023)), Electricity (Lai et al., 2018). These datasets exhibit strong channel heterogeneity, where sensors capture diverse spatial relationships and dynamic patterns, making them ideal for evaluating MTSF modeling capabilities. For scalability analysis, we additionally include the larger LargeST subsets GBA (2,352 channels), GLA (3,834 channels), and CA (8,600 channels). We prioritize traffic datasets because they exhibit strong cross-channel dependencies and *dynamic* sensor networks in practice, while datasets with weaker channel heterogeneity can be more efficiently handled by CI models (Shao et al., 2025b). A complete summary of dataset scales and split protocols follows the BasicTS+ (Shao et al., 2025b) standard (see Appendix Table 7).

**Evaluation metrics.** We employ two standard metrics to assess forecasting performance: Mean Absolute Error (MAE) and Weighted Absolute Percentage Error (WAPE). MAE provides a direct measure of prediction error in the original scale of the data, while WAPE normalizes across datasets with distributional shifts and scale disparities, enabling fair cross-dataset comparison (Liang et al., 2023).

**Baselines.** We compare CPiRi against a comprehensive suite of CI and CD models. **CI models:** DLinear (Zeng et al., 2023), PatchTST (Nie et al., 2023), Chronos-Bolt (Ansari et al., 2024), and Sundial (Liu et al., 2025). **CD models:** Informer (Zhou et al., 2021), STID (Shao et al., 2022), Crossformer (Zhang & Yan, 2023), iTransformer (Liu et al., 2024a), CrossGNN (Huang et al., 2023), TimeXer (Wang et al., 2024), and Timer-XL (Liu et al., 2024b).

**Implementation details.** All models are evaluated on the benchmark BasicTS+ (Shao et al., 2025b) with five separate training runs on a server equipped with NVIDIA A800 GPU, and adopt the optimal configurations. CPiRi's dropout rate is set to 0.3 for better constructing sparse spatial relation. Both $L$ and $T$ are 336 for all experiments. *More implementation details can be referred to appendix.*

## 4.2 COMPARISON WITH SOTA METHODS

**Forecasting accuracy.** As shown in Table 1, CPiRi achieves SOTA performance on four of five benchmark datasets under standard training protocols and significantly outperforms existing CI and CD models. The sole exception is METR-LA, where STID and Crossformer leverages exogenous holiday features unavailable to our sequence-only model. Crucially, CPiRi surpasses all large pre-trained baselines by substantial margins (>12% WAPE on SD), validating that our decoupled design better handles limited data and weak spatial signals than monolithic architectures.

Table 1: Main forecasting performance comparison. All models are trained with fixed channel order (except where noted), using official pre-trained weights (*) when full training was infeasible. CPiRi achieves SOTA performance while maintaining CPI, demonstrating the advantages of our hybrid approach. Best results are in **bold**; second-best are underlined.

| Paradigm | Model | METR-LA | | PEMS-BAY | | PEMS-04 | | PEMS-08 | | SD | | Electricity | |
|---|---|---|---|---|---|---|---|---|---|---|---|---|---|
| | | WAPE↓ | MAE↓ | WAPE↓ | MAE↓ | WAPE↓ | MAE↓ | WAPE↓ | MAE↓ | WAPE↓ | MAE↓ | WAPE↓ | MAE↓ |
| CI | Chronos-Bolt* | 24.19% | 11.76 | 8.52% | 5.11 | 34.48% | 73.97 | 32.83% | 71.76 | 19.71% | 44.11 | 10.22% | 241.44 |
| | Sundial* | 16.51% | 8.14 | 5.79% | 3.52 | 18.77% | 36.44 | 22.69% | 30.05 | 24.40% | 53.94 | 11.81% | 276.72 |
| | Dlinear | 14.93% | 7.67 | 5.22% | 3.17 | 17.77% | 36.92 | 16.50% | 33.42 | 19.46% | 43.93 | 11.80% | 244.86 |
| | PatchTST | 10.51% | 5.33 | 4.87% | 2.95 | 15.54% | 32.32 | 12.37% | 23.83 | 13.41% | 29.06 | 10.68% | 241.32 |
| CD | Timer-XL* | 23.64% | 12.27 | 8.22% | 4.95 | 36.33% | 77.05 | 31.52% | 68.07 | 46.07% | 110.47 | 17.23% | 405.63 |
| | Informer | 10.36% | 5.25 | 4.47% | 2.70 | 13.57% | 27.88 | 13.02% | 27.36 | 19.55% | 43.10 | 17.61% | 347.94 |
| | CrossGNN | 12.82% | 6.52 | 5.19% | 3.15 | 17.90% | 37.16 | 16.83% | 33.81 | 19.51% | 44.00 | 11.63% | 258.94 |
| | TimeXer | 11.18% | 5.69 | 4.61% | 2.79 | 16.43% | 34.16 | 16.02% | 31.66 | 14.43% | 31.37 | 18.70% | 357.05 |
| | iTransformer | 11.28% | 5.71 | 4.21% | 2.55 | 12.99% | 26.79 | 10.70% | 20.17 | 12.45% | 27.28 | 10.67% | 237.59 |
| | STID | **8.48%** | **4.21** | 3.91% | 2.36 | 12.43% | 25.65 | 10.90% | 20.60 | 12.51% | 26.64 | 10.65% | 245.24 |
| | Crossformer | 8.84% | 4.42 | 4.07% | 2.47 | 13.28% | 27.30 | 11.43% | 22.03 | 12.50% | 27.21 | 13.57% | 257.69 |
| CI+CD | **CPiRi** (ours) | 9.14% | 4.62 | **3.90%** | **2.36** | **11.67%** | **23.96** | **9.43%** | **17.46** | **12.25%** | 26.85 | **9.90%** | **235.33** |

Table 2: Channel shuffling robustness analysis. For each model, we show: (1) performance when trained normally but tested with shuffled channels (Test Shuffle), and (2) performance when trained with channel shuffling (Train Shuffle). The severe degradation of most CD models under test-time shuffling reveals their dependence on fixed channel order, while CPiRi maintains stable performance in all conditions.

| Model | Shuffle | METR-LA | | PEMS-BAY | | PEMS-04 | | PEMS-08 | | SD | |
|---|---|---|---|---|---|---|---|---|---|---|---|
| | | WAPE↓ | MAE↓ | WAPE↓ | MAE↓ | WAPE↓ | MAE↓ | WAPE↓ | MAE↓ | WAPE↓ | MAE↓ |
| Informer | Test | 20.19% | 10.38 | 9.99% | 6.03 | 83.53% | 150.02 | 118.19% | 145.58 | 19.55% | 43.10 |
| | Train | 16.48% | 8.11 | 7.30% | 4.38 | 49.39% | 90.51 | 74.00% | 81.12 | 17.38% | 38.15 |
| CrossGNN | Test | 12.85% | 6.53 | 5.20% | 3.15 | 17.95% | 37.23 | 17.01% | 33.89 | 19.51% | 44.00 |
| | Train | 12.78% | 6.48 | 5.19% | 3.15 | 17.92% | 37.18 | 16.79% | 33.78 | 19.49% | 43.93 |
| TimeXer | Test | 13.79% | 7.08 | 5.92% | 3.57 | 17.22% | 35.80 | 16.74% | 33.27 | 18.46% | 40.27 |
| | Train | 11.84% | 6.09 | 4.86% | 2.95 | 16.72% | 34.94 | 15.96% | 31.88 | 14.77% | 32.38 |
| iTransformer | Test | 11.27% | 5.70 | 4.21% | 2.55 | 12.99% | 26.79 | 10.70% | 20.17 | 12.45% | 27.28 |
| | Train | 11.50% | 5.86 | 4.21% | 2.55 | 13.02% | 26.84 | 10.58% | 19.98 | 12.40% | 27.21 |
| STID | Test | 18.07% | 9.23 | 7.20% | 4.35 | 52.31% | 86.25 | 65.18% | 69.20 | 12.51% | 26.64 |
| | Train | 10.11% | 5.15 | 4.30% | 2.60 | 13.75% | 28.17 | 11.82% | 21.93 | 12.98% | 28.18 |
| Crossformer | Test | 18.06% | 9.12 | 6.66% | 4.03 | 43.83% | 78.36 | 39.85% | 54.72 | 12.50% | 27.21 |
| | Train | 9.87% | 4.90 | 4.47% | 2.69 | 14.75% | 30.38 | 12.82% | 23.57 | 12.85% | 27.65 |
| **CPiRi** (ours) | Test | 9.23% | 4.67 | 4.02% | 2.45 | 11.93% | 24.57 | 10.08% | 18.20 | 13.46% | 29.21 |
| | Train | **9.14%** | **4.62** | **3.90%** | **2.36** | **11.67%** | **23.96** | **9.43%** | **17.46** | **12.25%** | **26.85** |

Table 3: Performance degradation under partial channel shuffling on the PEMS-08 dataset. The performance of non-invariant CD models collapses progressively as the percentage of permuted channels increases at test time, while CPiRi remains perfectly robust across all conditions. The best results are highlighted in **bold**, while the second-best results are underlined.

| Model | 100% shuffle | | 75% shuffle | | 50% shuffle | | 25% shuffle | | 0% shuffle | |
|---|---|---|---|---|---|---|---|---|---|---|
| | WAPE↓ | MAE↓ | WAPE↓ | MAE↓ | WAPE↓ | MAE↓ | WAPE↓ | MAE↓ | WAPE↓ | MAE↓ |
| Informer | 118.19% | 145.58 | 76.21% | 113.26 | 52.73% | 83.18 | 41.25% | 59.07 | 13.02% | 27.36 |
| STID | 65.18% | 69.20 | 42.66% | 56.13 | 30.17% | 41.31 | 25.61% | 32.19 | 10.90% | 20.60 |
| Crossformer | 39.85% | 54.72 | 28.33% | 46.86 | 22.29% | 38.41 | 19.90% | 33.06 | 11.43% | 22.03 |
| Timer-XL | 31.52% | 68.11 | 31.52% | 68.07 | 31.54% | 68.11 | 31.53% | 68.11 | 31.52% | 68.07 |
| CrossGNN | 17.01% | 33.89 | 16.91% | 33.86 | 16.86% | 33.84 | 16.89% | 33.85 | 16.83% | 33.81 |
| TimeXer | 16.74% | 33.27 | 16.57% | 32.97 | 16.40% | 32.63 | 16.19% | 32.06 | 16.02% | 31.66 |
| iTransformer | 10.70% | 20.17 | 10.70% | 20.17 | 10.70% | 20.17 | 10.70% | 20.17 | 10.70% | 20.17 |
| **CPiRi** (ours) | **9.43%** | **17.46** | **9.43%** | **17.46** | **9.43%** | **17.46** | **9.43%** | **17.46** | **9.43%** | **17.46** |

**Channel shuffling vulnerability analysis.** Table 2 exposes critical limitations in mainstream CD models when subjected to channel shuffling at inference time. Models trained without permutation augmentation exhibit catastrophic performance degradation under channel-shuffled testing exemplified by Informer and STID, representing error increases exceeding 400% and 235% respectively. This fragility stems from architectural dependencies on channel order, such as fixed positional encodings that incentivize index memorization over content-based reasoning.

**CPiRi's robustness advantage.** CPiRi sustains stable performance under both training and testing permutations (Table 2), with minimal deviation between standard and shuffled evaluations ($\Delta$WAPE $< 0.25\%$ across datasets). Unlike iTransformer, which achieves CPI by performing spatio-temporal multi-head attention inside each layer and thus couples temporal and channel dimensions with an approximate $O((T \times C)^2)$ cost, CPiRi fully decouples them: a frozen temporal encoder summarizes each channel into a last token representation, and a single $O(C^2)$ spatial attention then operates on these summaries. This design delivers stronger robustness and higher efficiency. *Full results with standard deviations are provided in appendix.*

**Progressive permutation robustness.** Table 3 quantifies model degradation under increasing permutation intensities on PEMS-08, where *25% shuffle* indicates sampling 25% channels uniformly at random and applying a random permutation within this subset, while keeping the remaining channels in their original order. CD models (e.g., Informer, STID) exhibit progressive performance collapse as shuffling rates increase, with WAPE deteriorating by $>100\%$ under full permutation.

Table 4: Ablation study of the CPiRi framework about the effectiveness of its core principles and architectural components.

| Variant | METR-LA | | PEMS-BAY | | PEMS-04 | | PEMS-08 | | SD | |
|---|---|---|---|---|---|---|---|---|---|---|
| | WAPE↓ | MAE↓ | WAPE↓ | MAE↓ | WAPE↓ | MAE↓ | WAPE↓ | MAE↓ | WAPE↓ | MAE↓ |
| CPiRi | 9.14% | 4.62 | 3.90% | 2.36 | 11.67% | 23.96 | 9.43% | 17.46 | 12.25% | 26.85 |
| w/o spatial-temporal decouple | 9.21% | 4.65 | 3.93% | 2.38 | 11.91% | 24.46 | 10.80% | 18.99 | 13.37% | 28.74 |
| w/o regularization strategy | 9.23% | 4.67 | 4.02% | 2.45 | 11.93% | 24.57 | 10.08% | 18.20 | 13.46% | 29.21 |
| w/o pretrained weights | 16.96% | 8.91 | 7.54% | 4.54 | 74.86% | 90.45 | 52.29% | 101.69 | 64.84% | 120.22 |
| w/ 3 layer encoder from scratch | 10.71% | 5.41 | 4.28% | 2.56 | 12.61% | 26.29 | 11.17% | 21.96 | 14.34% | 30.39 |
| w/ frozen Chronos-2 encoder | 10.90% | 5.47 | 4.32% | 2.66 | 16.54% | 34.16 | 13.16% | 25.66 | 18.50% | 40.39 |
| w/ fine-tuning in last 10 epochs | **8.81%** | **4.46** | 4.00% | 2.42 | 11.86% | 24.35 | 9.73% | 17.82 | **12.00%** | **26.47** |
| w/o spatial module | 16.51% | 8.14 | 5.79% | 3.52 | 18.77% | 36.44 | 22.69% | 30.05 | 24.40% | 53.94 |
| w/ mean pooling | 13.72% | 7.07 | 4.24% | 2.56 | 12.81% | 26.21 | 12.42% | 21.89 | 13.33% | 29.03 |

In stark contrast, CPiRi maintains invariant prediction quality regardless of permutation intensity (9.43% WAPE at all shuffling levels). This establishes channel shuffling tests as essential diagnostics for dynamic real-world deployments where sensor configurations frequently change. Across datasets with different channel counts and under progressive shuffling, the degradation pattern aligns more with the strength of inter-channel correlations than with channel counts itself, indicating that brittleness under reordering is driven primarily by channel heterogeneity rather than the number of channels.

## 4.3 Ablation Study

**Framework components.** We first validate our two primary contributions, with results summarized in Table 4. In the *w/o spatial-temporal decouple* variant, we fine-tune the encoder alongside the spatial module instead of keeping it frozen. This leads to performance degradation, confirming our hypothesis that decoupling is crucial. It prevents the model from overfitting to dataset-specific artifacts and preserves the robust priors from the foundation model. The *w/o regularization strategy* variant, which disables the channel shuffling strategy, shows a consistent performance drop. This demonstrates that our regularization strategy is a potent regularizer, compelling the model to learn truly generalizable, content-driven relationships rather than relying on positional shortcuts.

**Encoder variants and fine-tuning.** Beyond the core ablations, Table 4 further clarifies how the temporal encoder choice and fine-tuning strategy interact with our spatio-temporal decoupling and CPI objective. First, *w/o pretrained weights* simply removes Sundial pretraining while keeping the same architecture; this variant collapses, indicating that high-quality temporal priors are indispensable for our content-driven relational reasoning in the spatial module. Second, *w/ 3 layer encoder from scratch* replaces the original 12-layer Sundial encoder with a lightweight 3-layer counterpart trained from random initialization. Although this variant converges more readily than *w/o pretrained weights*, it still lags behind CPiRi, underscoring that a strong frozen encoder is key to our decoupled design. Third, *w/ frozen Chronos-2 encoder* uses Chronos-2 (Ansari et al., 2025) with 120M parameters as encoder to replace Sundial with 128M parameters. Since Chronos series models are designed around short forecasting horizons of 64, their representations transfer poorly to our long-horizon setting, and the performance is even inferior to the 3-layer-from-scratch variant. Finally, *w/ fine-tuning in last 10 epochs* unfreezes the encoder in last 10 epochs. While this brings slight gains on a few datasets, it raises training memory consumption by about five times and reduces representation separability in UMAP analyses (Fig. 6), suggesting overfitting to a single dataset and weakening the CPI-oriented decoupling. Overall, these findings support our design choice: coupling a *frozen* pretrained temporal encoder with a *permutation-equivariant* spatial block trained under channel shuffling best balances accuracy, robustness under co-drift, and practical efficiency.

**Model designs.** We also ablate the key modules of our model architecture (Table 4). Removing the spatial module (*w/o spatial module*) causes a catastrophic drop in performance, reducing the model to a simple CI forecaster. This irrefutably proves that explicit cross-channel relational modeling is essential for achieving SOTA accuracy. Specifically, relying on the final token for next-token prediction is well-suited forecasting; replacing this with an aggregated average of all tokens results in a notable performance drop *w/ mean pooling*, further validating our design's effectiveness.

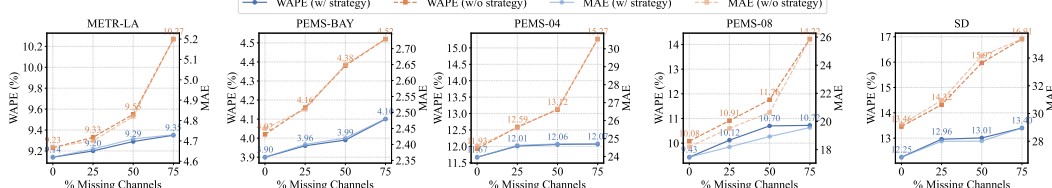

Figure 4: Analysis of inductive generalization to unseen channels and the impact of permutation-invariant regularization strategy. Performance is evaluated with and without the strategy as the percentage of available training channels varies. The results show that the strategy consistently improves performance, with the benefits becoming more pronounced in low-data regimes. For clarity, WAPE (%) values are annotated directly on the corresponding data points.

Table 5: Forecasting performance on large-scale datasets.

| Dataset | Channels | Timer-XL | | Sundial | | CPiRi | |
|---|---|---|---|---|---|---|---|
| | | WAPE↓ | MAE↓ | WAPE↓ | MAE↓ | WAPE↓ | MAE↓ |
| SD | 716 | 46.07% | 110.47 | 24.40% | 53.94 | **12.25**% | **26.85** |
| GBA | 2,352 | 42.99% | 100.10 | 21.02% | 48.08 | **11.48**% | **25.50** |
| GLA | 3,834 | 40.83% | 106.15 | 23.21% | 54.46 | **11.00**% | **26.40** |
| CA | 8,600 | 42.75% | 96.53 | 23.60% | 49.56 | **12.68**% | **25.94** |

Table 6: Efficiency and scalability analysis on the large-scale CA dataset ($C$=8600, $T$=336). The comparison highlights the practical advantages of CPiRi's decoupled architecture over monolithic models like Timer-XL, which are often constrained by prohibitive memory requirements. "OOM" denotes an out-of-memory error.

| Paradigm | Model | Maximum batch size | Inference time (s) | | Avg. time (s) | | GPU memory (GB) | | Avg. memory (GB) | | Complexity |
|---|---|---|---|---|---|---|---|---|---|---|---|
| | | | Base | Compiled | Base | Compiled | Base | Compiled | Base | Compiled | |
| CI | Sundial | 4 | 2.55 | 1.61 | 0.64 | 0.40 | 54.44 | 20.68 | 13.61 | 5.17 | $O(T^2)$ |
| CI+CD | **CPiRi** (ours) | 4 | 2.66 | 1.62 | 0.67 | 0.41 | 54.56 | 32.00 | 13.64 | 8.00 | $O(T^2 + C^2)$ |
| CD | Timer-XL | 1 | OOM | 1.07 | – | 1.07 | >80 | 75.68 | – | 75.68 | $O((T \times C)^2)$ |
| CD | iTransformer | 2 | 0.52 | 0.40 | 0.26 | 0.20 | 45.18 | 35.37 | 22.59 | 17.69 | $O((T \times C)^2)$ |

**Generalization under data scarcity.** To further probe the limits of CPiRi, we test its performance on the full dataset when trained on only a fraction of the available channels (25%, 50%, 75%). As illustrated in Fig. 4, CPiRi demonstrates remarkable inductive generalization to unseen channels. Notably, training with only 25% of channels reduces training time by 70% while incurring merely a 2% drop in accuracy, further highlighting the efficiency of our framework in resource-constrained scenarios. CPiRi maintains competitive performance when trained on 50% of channels, demonstrating its capability for handling unseen channels based on temporal patterns. The widening performance gap in low-data regimes confirms that our regularization strategy is critical for robust and generalizable relational reasoning under challenging, data-scarce conditions.

## 4.4 EXPLORATORY STUDY

**Scalability and efficiency analysis.** CPiRi delivers consistent gains on large-scale benchmarks (up to 8,600 channels; Table 5), confirming real-world scalability while keeping runtime and memory modest. The decoupled design has complexity ($O(C^2 + T^2)$): temporal encoding and decoding are handled by the frozen CI backbone, and the spatial module attends only across the $C$ last tokens per channel rather than over all tokens, which is far more tractable than Timer-XL's $O((T \times C)^2)$. On the CA with 8,600 channels, Table 6 reports compiled per-instance inference of 0.41s for CPiRi versus Sundial (0.40s), with average GPU memory 8.00GB versus 5.17GB. Timer-XL demands 75.68GB memory under comparable settings. CPiRi requires substantially less memory compared with iTransformer, which performs multi-layer spatiotemporal aggregation inside each attention block and raises computational cost and makes optimization harder. This confirms CPiRi strikes an effective balance between modeling capacity and computational cost, making it a practical solution for large-scale tasks.

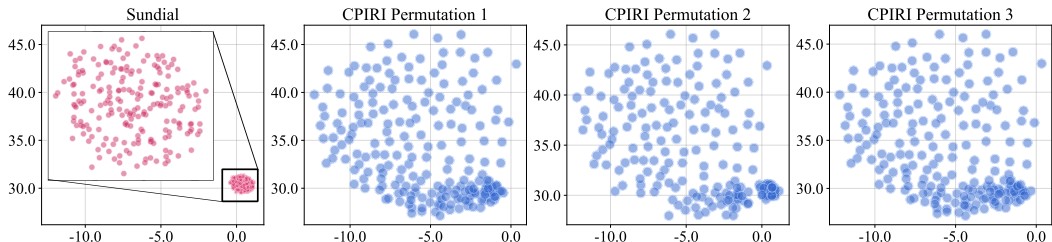

Figure 5: UMAP visualization of channel representations on METR-LA. Left: Sundial. Right: CPiRi under three random channel permutations, near-identical geometry with clearer separation.

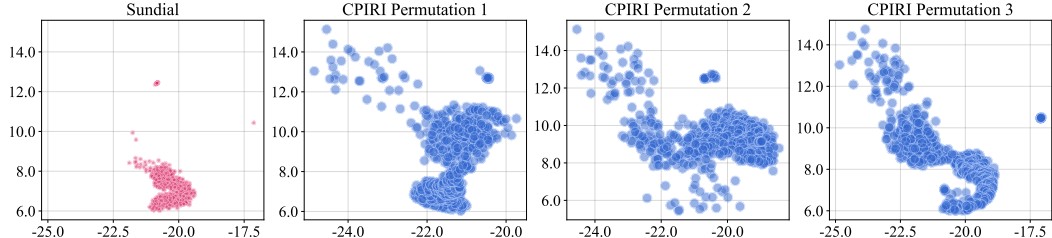

Figure 6: UMAP visualization of channel representations on METR-LA after fine-tuning CPiRi, embeddings become more compact and less separable.

**Qualitative analysis.** We use UMAP (Ghojogh et al., 2023) to visualize channel representations before and after the spatial module. Fig. 5 contains four panels: the left panel shows raw Sundial embeddings, and the three right panels show CPiRi outputs under three independently shuffled channel orders, all projected into the same space. The three CPiRi clouds exhibit nearly identical geometry and clearly improved cluster separability relative to Sundial, illustrating that the permutation-equivariant spatial attention plus channel-shuffling regularization learns content-driven, order-agnostic relationships consistent with CPI. Fig. 6 visualizes embeddings after late-stage unfreezing of the encoder. Compared with the frozen-encoder CPiRi, the points become overly compact and less separable across channels, suggesting partial representation collapse and dataset-specific overfitting. Together with Appendix Fig. 9 across datasets, the embeddings after CPiRi exhibit a more semantically meaningful structure with clearer separability across channels, consistent with the higher-fidelity forecasts in the appendix case studies (Fig. 10).

## 5 CONCLUSION

This paper proposes CPiRi, a framework that successfully synergizes the strengths of the CI and CD paradigms. Its design is founded on two principles: a radical spatio-temporal decoupling and a permutation-invariant regularization strategy. The architecture leverages a frozen univariate foundation model to provide robust temporal features, which are then processed by a lightweight spatial module. The strategy guided by CPI forces spatial module to learn content-driven relational reasoning meta-skill, which not only enhances data diversity but also improves the channel discriminability. This approach leads to SOTA predictive accuracy on multiple benchmarks, and overcomes the brittleness of complex CD models that overfit to positional artifacts, a critical flaw revealed by the channel shuffling test. Critically, CPiRi exhibits: (1) strong inductive generalization to unseen channels (e.g., 50% training channels), and (2) enhanced robustness in low-data regimes, where its regularization strategy prevents overfitting to sparse relational patterns. Beyond accuracy and robustness, CPiRi is practical at scale. These properties make CPiRi a compelling choice for real-world environments with structural and distributional co-drift, where relational reasoning and operational efficiency are both required.

**Limitation discussion.** While CPiRi synergizes CI/CD strengths effectively, its static fusion mechanism between temporal and spatial modules presents a limitation for scenarios involving abrupt trend shifts. Future work will pursue dynamic fusion mechanisms by developing adaptive interaction protocols, potentially improving responsiveness to volatile patterns. A more ambitious frontier is to move beyond purely endogenous signals by integrating external, unstructured information (e.g., news events, policy changes) within a causal reasoning framework to move beyond endogenous modeling, enhancing real-world applicability.

ACKNOWLEDGMENTS

This work has been supported by Zhejiang Province Science and Technology Leading Talent Plan Project of China (No.2023R5213), Zhejiang Province "Lingyan" Key R&D Project of China (No.2026C02A2002), Zhejiang Province "Jianbing" Key R&D Project of China (No.2025C01010, No.2024C01034, No.2025C01144), and Zhejiang Provincial Natural Science Foundation of China (No.LQN26F030003).

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

# A    APPENDIX

## A.1    MORE EXPERIMENTAL SETUP

### A.1.1    DATASET SELECTION AND PROTOCOLS

**Rationale for dataset selection.** A rigorous evaluation of MTSF models hinges on datasets with significant *channel heterogeneity* (Shao et al., 2025b), where individual time series exhibit diverse and complex dynamics. Such datasets are essential for assessing a model's ability to learn meaningful inter-channel relationships. We specifically select traffic forecasting benchmarks as they provide an ideal testbed for two reasons. First, they are known for their strong and complex inter-channel dependencies. Second, they are derived from dynamic sensor networks where events like sensor failures or expansions are common, making Channel Permutation Invariance (CPI) a practical necessity for robust real-world deployment.

Table 7: Statistics of the datasets used in our experiments.

| Dataset | Channels | Timesteps | Granularity | Time Period | Data Type | Source |
|---------|----------|-----------|-------------|-------------|-----------|--------|
| METR-LA | 207 | 34,272 | 5 mins | Mar-Jun 2012 | Traffic Speed | Li et al. (2018) |
| PEMS-BAY | 325 | 52,116 | 5 mins | Jan-May 2017 | Traffic Speed | Li et al. (2023) |
| PEMS-04 | 307 | 16,992 | 5 mins | Jan-Feb 2018 | Traffic Flow | Chen et al. (2001) |
| PEMS-08 | 170 | 17,856 | 5 mins | Jul-Aug 2016 | Traffic Flow | Chen et al. (2001) |
| SD | 716 | 525,888 | 5 mins | 2017 - 2021 | Traffic Speed | Liu et al. (2023) |
| GBA | 2,352 | 525,888 | 5 mins | 2017 - 2021 | Traffic Speed | Liu et al. (2023) |
| GLA | 3,834 | 525,888 | 5 mins | 2017 - 2021 | Traffic Speed | Liu et al. (2023) |
| CA | 8,600 | 525,888 | 5 mins | 2017 - 2021 | Traffic Speed | Liu et al. (2023) |
| Electricity | 321 | 26,304 | 60 mins | 2012 - 2014 | Electricity Consumption | Lai et al. (2018) |

**Benchmark summary.** Our evaluation is conducted on a hierarchy of six standard benchmarks (METR-LA, PEMS-BAY, PEMS-04, PEMS-08, SD, Electricity) and is further extended to three large-scale datasets from LargeST (GBA, GLA, CA) to assess scalability. Detailed statistics for all datasets are summarized in Table 7. To ensure fair and reproducible comparisons, all dataset processing, including normalization and data splits, strictly adheres to the standardized protocols established by the BasicTS+ open-source benchmarking framework (Shao et al., 2025b).

### A.1.2    MODEL CONFIGURATIONS AND BASELINES

Our model, CPiRi, is built upon a frozen, pre-trained Sundial (Liu et al., 2025) model which serves as the backbone for temporal feature extraction. A lightweight, trainable Transformer encoder block functions as the spatial module, tasked with learning content-driven relational interactions. This design allows CPiRi to leverage powerful temporal priors while focusing training exclusively on the relational learning skill. Key hyperparameters, such as a dropout rate of 0.3, are kept consistent across all datasets to ensure robustness and minimize tuning overhead, as detailed in Table 8 and Fig. 7.

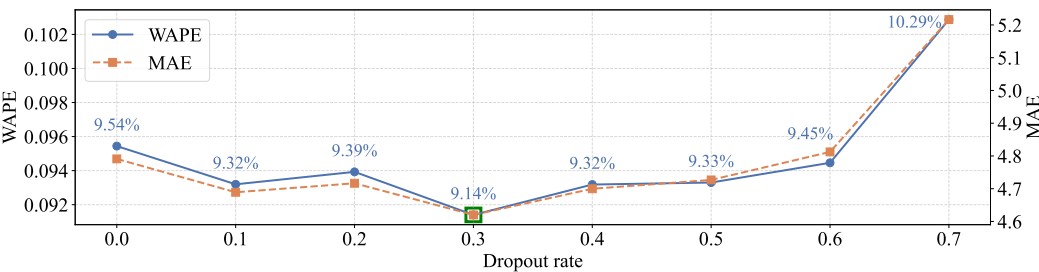

Figure 7: Hyperparameter sensitivity analysis of the dropout rate on the METR-LA dataset. The results indicate that a dropout rate of 0.3 achieves the optimal balance for both WAPE and MAE metrics. For clarity, WAPE values are annotated directly on the corresponding data points, and the best-performing setting is marked with a green square.

Table 8: Model Hyperparameters.

| Hyperparameter | Value |
|---|---|
| Input sequence length | 336 |
| Output sequence length | 336 |
| Number of layers | 4 |
| Learning rate (LR) | 0.001 |
| LR milestones | 1, 10, 25, 40 |
| Weight decay | $1 \times 10^{-5}$ |
| Gradient clipping | 3.0 |
| Dropout rate | 0.3 |
| Hidden dimension | 768 |
| Attention heads | 12 |
| Normalization | LayerNorm |
| Activation function | GELU |

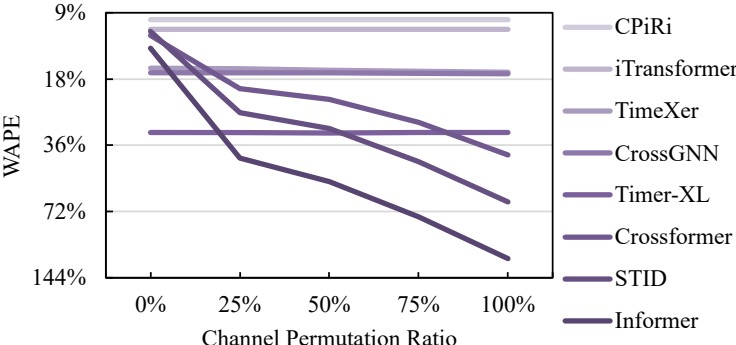

Figure 8: Performance degradation of CD models under partial channel shuffling on the PEMS-08 dataset. Models were trained with a fixed channel order. The performance of non-invariant models collapses progressively as the percentage of permuted channels increases at test time. In contrast, CPiRi and iTransformer remain perfectly robust across all conditions.

For a comprehensive comparison, all baseline models were evaluated within the BasicTS+ framework (Shao et al., 2025b) using their officially reported optimal hyperparameters. Our baseline selection spans the current MTSF landscape: we include channel-independent (CI) models like DLinear (Zeng et al., 2023) and PatchTST (Nie et al., 2023), which achieve permutation invariance by forgoing cross-channel interaction. These are contrasted with classic channel-dependent (CD) models, such as Informer (Zhou et al., 2021), STID (Shao et al., 2022), and Crossformer (Zhang & Yan, 2023), that explicitly model these relationships but, due to architectural biases like fixed positional encodings, prove brittle under permutation tests (Fig. 8). As a direct competitor, we include the CPI-aware iTransformer (Liu et al., 2024a), which achieves invariance through an innovative architectural shift. Finally, to benchmark data and parameter efficiency, we compare against large foundation models like Chronos-Bolt (Ansari et al., 2024) and Timer-XL (Liu et al., 2024b) using their official pre-trained weights.

### A.1.3 SCOPE: DYNAMIC MTSF AND EVALUATION PROTOCOL

Our experimental design is driven by the goal of *dynamic* MTSF rather than generic TSF. In deployment, channel configurations (ordering, availability) and spatial relations may change over time, and datasets exhibit strong *channel heterogeneity* and inter-channel dependence. Consequently, we prioritize (i) benchmarks where cross-channel structure matters and (ii) diagnostics that directly probe relational robustness (e.g., CPI tests). To make "dynamic" precise, we use structural and distributional co-drift to denote deployments where the channel set or topology changes (structural) and the underlying data distributions shift over time (distributional). Such co-drift is common in sensor networks, finance, and other operational settings.

**Metric rationale (WAPE vs. MSE/RMSE).** Because datasets differ in magnitude and units, we adopt WAPE as a scale-independent primary metric and report MAE in parallel for interpretability. Formally,

$$\text{WAPE} = \frac{\sum_t |y_t - \hat{y}_t|}{\sum_t |y_t| + \varepsilon},$$

which normalizes absolute error by the aggregate signal scale and yields a unit-free percentage that is directly comparable across datasets with different units/ranges. In contrast, MSE/RMSE are *scale-dependent*: their values grow with the variance and units of the target, making cross-dataset comparisons misleading and biasing summaries toward high-magnitude series. WAPE also avoids the divide-by-zero instability of MAPE when $y_t = 0$ by using the dataset-level denominator; we follow common practice with a small $\varepsilon$ (e.g., $10^{-8} \times$ mean $|y_t|$) for numerical safety. This protocol emphasizes adaptability and comparability across heterogeneous settings, instead of strictly mirroring generic TSF test beds that assume static channel layouts.

### A.1.4 ON CHANNEL ADDITION/REMOVAL EVALUATION

We operationalize "unseen channels" by training on subsets of channels (25%, 50%, 75%) and evaluating on the full set (Fig. 4). This protocol stresses inductive generalization without retraining and is agnostic to model families. In contrast, a *direct* plug-and-play test of channel addition/removal is not uniformly applicable to many CD baselines: their architectures bind the channel dimension via learned adjacency/positional structures, so adding or removing channels typically requires graph re-specification and (partial) retraining. Forcing comparators into this setting via ad-hoc workarounds (e.g., zero-masking channels to emulate missing/new inputs, as done in prior work such as YOAO (Zhang et al., 2025)) changes their effective training distribution and creates fairness issues across methods. For these reasons, we (1) use subset-training/full-test to approximate addition and removal in a model-agnostic manner, (2) use permutation (shuffle) tests to isolate order sensitivity, and (3) leave a comprehensive, architecture-specific add/remove benchmark, requiring consistent reparameterization rules for CD models, as future work. Notably, CPiRi's decoupled, permutation-equivariant design is inherently compatible with plug-and-play channel changes without retraining; the above constraints arise from ensuring fair cross-paradigm comparison rather than from limitations of CPiRi itself.

### A.1.5 COMPUTING WORKSTATION

All experiments were conducted on a server with the following specifications to ensure reproducibility:

- GPU: NVIDIA A800 (80GB)
- CPU: 128 Intel Xeon Platinum 8358 CPU @ 2.60GHz
- Operating System: Ubuntu 22.04 LTS
- Software: Python 3.10, PyTorch 2.3.1, CUDA 12.1

## A.2 EXTENDED EXPERIMENTAL RESULTS

### A.2.1 FORECASTING PERFORMANCE AND STABILITY

Table 9 reports the full forecasting results, extending the main paper by providing both the mean and standard deviation (STD) over five independent runs. The STD serves as a critical indicator of model stability and reliability. The results show that CPiRi not only achieves state-of-the-art (SOTA) accuracy but also consistently exhibits a smaller standard deviation across most datasets compared to complex baselines. This heightened stability is a direct consequence of its robust, regularized design, which combines a frozen pre-trained backbone with the channel shuffling strategy to prevent overfitting and enhance reproducibility.

### A.2.2 ON THE SOURCE OF CPiRi'S SUPERIORITY

The superiority of CPiRi stems from a powerful synergy between its permutation-equivariant architecture and its unique training methodology. We substantiate this claim through two key analyses:

Table 9: Full forecasting performance comparison with standard deviations. This table extends and integrates the results from Table 1 and Table 2 of the main paper, reporting both mean and standard deviation (STD) over five independent runs. We evaluate all models under three distinct conditions: (1) standard training with a **fixed channel order**, (2) training with **channel shuffling**, and (3) testing a fixed-order model with **channel shuffling**. CPiRi achieves SOTA performance while maintaining perfect Channel Permutation Invariance (CPI). Best results are in **bold**; second-best are underlined.

| Paradigm | Model | METR-LA | | PEMS-BAY | | PEMS-04 | | PEMS-08 | | SD | |
|---|---|---|---|---|---|---|---|---|---|---|---|
| | | WAPE | STD | WAPE | STD | WAPE | STD | WAPE | STD | WAPE | STD |
| CI | Chronos-Bolt* | 24.19% | – | 8.52% | – | 34.48% | – | 32.83% | – | 19.71% | – |
| | Sundial* | 16.51% | – | 5.79% | – | 18.77% | – | 22.69% | – | 24.40% | – |
| | Dlinear | 14.93% | 0.11 | 5.22% | 0.07 | 17.77% | 0.13 | 16.50% | 0.12 | 19.46% | 0.16 |
| | PatchTST | 10.51% | 0.09 | 4.87% | 0.06 | 15.54% | 0.11 | 12.37% | 0.09 | 13.41% | 0.12 |
| CD trained with **fixed** channel order | Timer-XL* | 23.64% | – | 8.22% | – | 36.33% | – | 31.52% | – | 46.07% | – |
| | Informer | 10.36% | 0.09 | 4.47% | 0.06 | 13.57% | 0.11 | 13.02% | 0.10 | 19.55% | 0.17 |
| | CrossGNN | 12.82% | 0.11 | 5.19% | 0.07 | 17.90% | 0.14 | 16.83% | 0.13 | 19.51% | 0.17 |
| | TimeXer | 11.18% | 0.10 | 4.61% | 0.06 | 16.43% | 0.13 | 16.02% | 0.12 | 14.43% | 0.13 |
| | iTransformer | 11.28% | 0.10 | 4.21% | 0.05 | 12.99% | 0.10 | 10.70% | 0.08 | 12.45% | 0.11 |
| | STID | **8.48**% | 0.07 | 3.91% | 0.05 | 12.43% | 0.10 | 10.90% | 0.09 | 12.51% | 0.11 |
| | Crossformer | 8.84% | 0.08 | 4.07% | 0.05 | 13.28% | 0.11 | 11.43% | 0.09 | 12.50% | 0.11 |
| CD **trained** with channel **shuffling** | Informer | 16.48% | 0.25 | 7.30% | 0.15 | 49.39% | 0.45 | 74.00% | 0.60 | 17.38% | 0.28 |
| | CrossGNN | 12.78% | 0.18 | 5.19% | 0.10 | 17.92% | 0.25 | 16.79% | 0.24 | 19.49% | 0.28 |
| | TimeXer | 11.84% | 0.17 | 4.86% | 0.09 | 16.72% | 0.23 | 15.96% | 0.22 | 14.77% | 0.21 |
| | iTransformer | 11.50% | 0.16 | 4.21% | 0.08 | 13.02% | 0.18 | 10.58% | 0.15 | 12.40% | 0.17 |
| | STID | 10.11% | 0.15 | 4.30% | 0.08 | 13.75% | 0.19 | 11.82% | 0.17 | 12.98% | 0.18 |
| | Crossformer | 9.87% | 0.14 | 4.47% | 0.08 | 14.75% | 0.20 | 12.82% | 0.18 | 12.85% | 0.18 |
| CD **tested** with channel **shuffling** | Informer | 20.19% | 0.30 | 9.99% | 0.20 | 83.53% | 0.70 | 118.19% | 0.90 | 19.55% | 0.35 |
| | CrossGNN | 12.85% | 0.20 | 5.20% | 0.12 | 17.95% | 0.28 | 17.01% | 0.26 | 19.51% | 0.30 |
| | TimeXer | 13.79% | 0.22 | 5.92% | 0.14 | 17.22% | 0.27 | 16.74% | 0.25 | 18.46% | 0.29 |
| | iTransformer | 11.27% | 0.18 | 4.21% | 0.10 | 12.99% | 0.20 | 10.70% | 0.17 | 12.45% | 0.19 |
| | STID | 18.07% | 0.28 | 7.20% | 0.16 | 52.31% | 0.50 | 65.18% | 0.55 | 12.51% | 0.20 |
| | Crossformer | 18.06% | 0.28 | 6.66% | 0.15 | 43.83% | 0.45 | 39.85% | 0.40 | 12.50% | 0.20 |
| CI+CD | **CPiRi** (ours) | 9.14% | 0.08 | **3.90**% | 0.05 | **11.67**% | 0.09 | **9.43**% | 0.07 | **12.25**% | 0.11 |

first, by evaluating its robustness and accuracy in permuted channel environments, and second, by demonstrating its generalization capability in data-scarce scenarios.

The channel permutation tests, detailed in Table 9 and visualized in Fig. 8, reveal a clear hierarchy of model capabilities. Architecturally biased models like Informer and STID fail catastrophically, and even when trained with channel shuffling, they cannot overcome their inherent limitations, leading to compromised performance. While iTransformer's purely architectural solution achieves perfect robustness, CPiRi consistently surpasses it in accuracy (e.g., 9.43% vs. 10.70% WAPE on PEMS-08). This highlights that CPiRi's combination of a robust architecture and a targeted training approach is more effective.

This principle, that the training process unlocks a deeper, more generalizable relational reasoning, is further validated by the model's performance in low-data regimes, presented in Table 10. When trained on progressively smaller subsets of channels, the performance gap between models trained with and without our channel shuffling strategy widens dramatically. For instance, on PEMS-08, the WAPE gap is minor when using all channels (9.43% vs. 10.08%), but expands significantly when trained on only 25% of channels (10.72% vs. 14.22%). This confirms that this regularization technique acts as a potent regularizer, compelling the model to learn a "meta-skill" of content-driven reasoning. This learned generalization is the primary source of CPiRi's superiority.

### A.2.3 STATISTICAL SIGNIFICANCE TESTS

To rigorously validate our claims, we conducted pairwise Wilcoxon signed-rank tests (Table 11). The analysis confirms that CPiRi's performance improvement is statistically significant ($p<0.05$) over all baseline models under the challenging channel shuffling settings. While this performance gap narrows in static, fixed-order environments against highly specialized models like STID, CPiRi's unique advantage is achieving competitive SOTA accuracy without compromising the robustness crucial for real-world dynamic deployments.

Table 10: Validation of the channel shuffling strategy and its impact on generalization. This table compares performance with and without the strategy under data-scarce conditions, where training is performed on reduced subsets of the available channels (e.g., 'w/o 75%' signifies training on a 25% subset). The widening performance gap in low-data regimes highlights the strategy's critical role as a regularizer for learning generalizable relationships.

| Variant | Channels | METR-LA | | PEMS-BAY | | PEMS-04 | | PEMS-08 | | SD | |
|---|---|---|---|---|---|---|---|---|---|---|---|
| | | WAPE↓ | MAE↓ | WAPE↓ | MAE↓ | WAPE↓ | MAE↓ | WAPE↓ | MAE↓ | WAPE↓ | MAE↓ |
| w/ strategy | w/o 75% | 9.35% | 4.73 | 4.10% | 2.48 | 12.07% | 24.72 | 10.72% | 19.57 | 13.40% | 28.96 |
| | w/o 50% | 9.29% | 4.71 | 3.99% | 2.42 | 12.06% | 24.70 | 10.70% | 18.93 | 13.01% | 28.01 |
| | w/o 25% | 9.20% | 4.66 | 3.96% | 2.40 | 12.01% | 24.62 | 10.12% | 18.19 | 12.96% | 28.00 |
| | – | 9.14% | 4.62 | 3.90% | 2.36 | 11.67% | 23.96 | 9.43% | 17.46 | 12.25% | 26.85 |
| w/o strategy | w/o 75% | 10.27% | 5.20 | 4.52% | 2.73 | 15.27% | 30.54 | 14.22% | 25.86 | 16.91% | 35.37 |
| | w/o 50% | 9.55% | 4.82 | 4.38% | 2.65 | 13.12% | 26.65 | 11.76% | 20.64 | 15.97% | 34.16 |
| | w/o 25% | 9.33% | 4.70 | 4.16% | 2.51 | 12.59% | 25.60 | 10.91% | 19.51 | 14.32% | 30.95 |
| | – | 9.23% | 4.67 | 4.02% | 2.45 | 11.93% | 24.57 | 10.08% | 18.20 | 13.46% | 29.21 |

Table 11: Significance test results of WAPE values using the Wilcoxon signed-rank test. Our model, **CPiRi**, is compared against other models across the 5 datasets. The p-values are for a one-tailed test with the alternative hypothesis that CPiRi's WAPE is lower. We use a significance level of $\alpha = 0.05$. Results where CPiRi is statistically superior (p $<$0.05) are marked in **bold**.

| Comparison Model | Test Statistic (W) | p-value | p $<$0.05 |
|---|---|---|---|
| *Paradigm: CI* | | | |
| Dlinear | 0.0 | **0.031** | Yes |
| PatchTST | 0.0 | **0.031** | Yes |
| *Paradigm: CD (trained with fixed channel order)* | | | |
| Informer | 0.0 | **0.031** | Yes |
| CrossGNN | 0.0 | **0.031** | Yes |
| TimeXer | 0.0 | **0.031** | Yes |
| iTransformer | 0.0 | **0.031** | Yes |
| STID | 3.0 | 0.156 | No |
| Crossformer | 3.0 | 0.156 | No |
| *Paradigm: CD (trained with channel shuffling)* | | | |
| Informer | 0.0 | **0.031** | Yes |
| CrossGNN | 0.0 | **0.031** | Yes |
| TimeXer | 0.0 | **0.031** | Yes |
| iTransformer | 0.0 | **0.031** | Yes |
| STID | 0.0 | **0.031** | Yes |
| Crossformer | 0.0 | **0.031** | Yes |
| *Paradigm: CD (tested with channel shuffling)* | | | |
| Informer | 0.0 | **0.031** | Yes |
| CrossGNN | 0.0 | **0.031** | Yes |
| TimeXer | 0.0 | **0.031** | Yes |
| iTransformer | 0.0 | **0.031** | Yes |
| STID | 0.0 | **0.031** | Yes |
| Crossformer | 0.0 | **0.031** | Yes |

## A.3 ADDITIONAL QUALITATIVE ANALYSIS

### A.3.1 VISUALIZATIONS OF CHANNEL REPRESENTATIONS

To provide intuitive evidence of what the spatial module learns, we use UMAP (Ghojogh et al., 2023) to visualize the channel representations before and after its application in Fig. 9 for all benchmark datasets, extending the analysis from the main paper. The results are remarkably consistent: the initial temporal representations from the Sundial encoder (red) form a dense, undifferentiated cluster. In stark contrast, after being processed by the CPiRi spatial module, the representations (blue) are organized into well-separated, structured manifolds. This clearly shows that the spatial module successfully learns to reconfigure the representations into a more semantically meaningful structure,

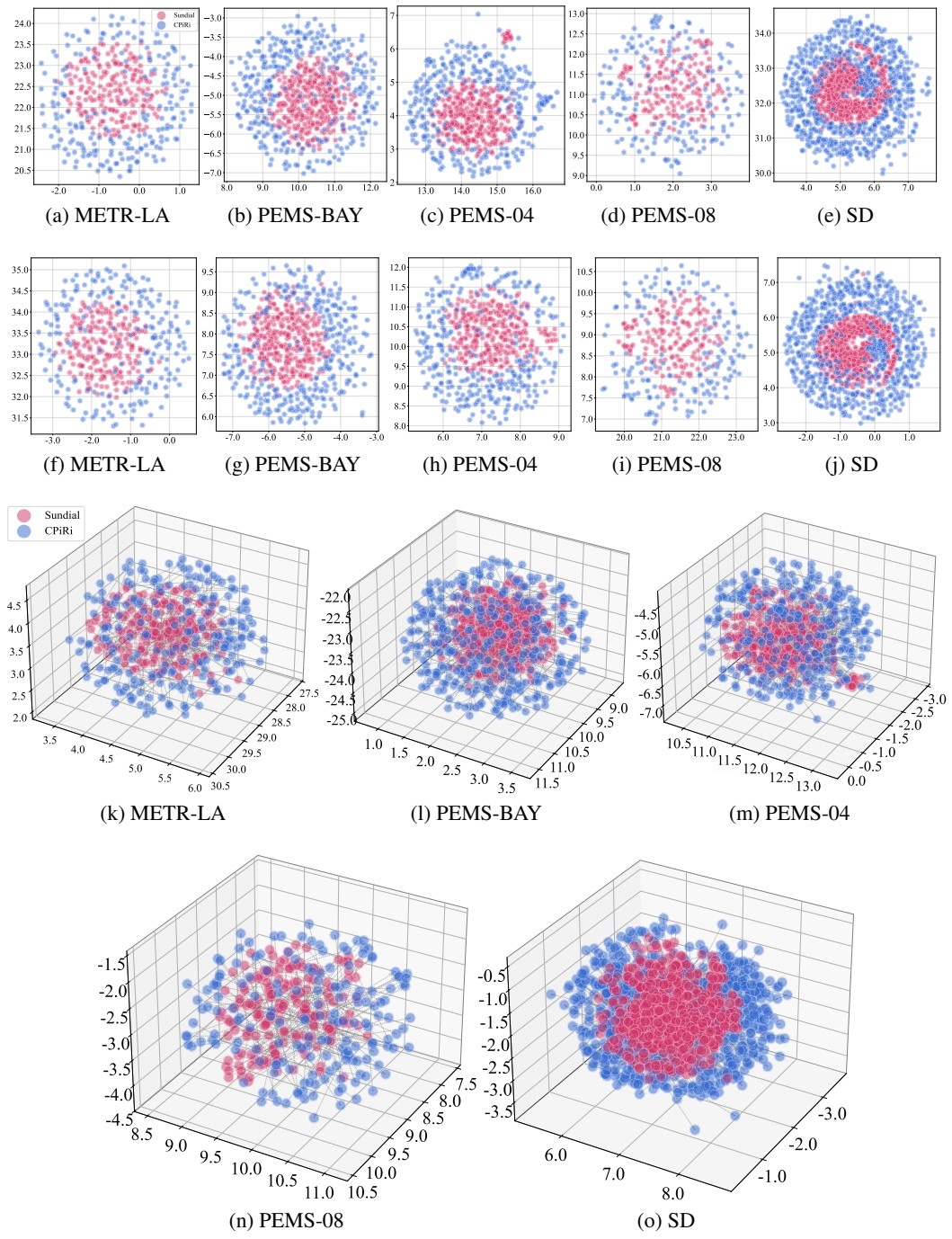

Figure 9: UMAP visualization of channel representations. Temporal feature distributions from the Sundial encoder (red) versus CPiRi-processed representations (blue) across datasets. CPiRi induces semantically meaningful clustering evidenced by the structured separation of blue manifolds, demonstrating its capacity for content-driven relational reasoning guides by CPI. Sub-figures (a-j) show two-dimensional projections. Sub-figures (k-o) show three-dimensional projections, where vectors illustrate the transformation from initial (Sundial) to final (CPiRi) representations.

improving their distinguishability. This consistent transformation provides powerful visual proof that CPiRi has learned a transferable "meta-skill" for identifying and encoding relational patterns based purely on content, transcending the specifics of any single dataset.

### A.3.2 CASE STUDY OF FORECASTING PERFORMANCE

To provide further intuitive evidence of CPiRi's effectiveness, Fig. 10 visualizes 336-step forecasting results for sample cases from the benchmark datasets. The plots clearly show that CPiRi's predictions maintain high fidelity to the ground truth across diverse patterns and time scales, demonstrating that the spatial interaction module successfully utilizes inter-channel dependencies to refine its forecasts. In contrast, the channel-independent Sundial baseline struggles to capture the unique dynamics of each channel, resulting in noticeable deviations. This qualitative comparison underscores the critical role of relational learning in achieving accurate multivariate forecasting.

### A.4 EXTENDED ANALYSIS

### A.4.1 THE STRUCTURAL COUPLING OF CHANNEL STRATEGY AND DECODING PARADIGM

The ongoing discourse regarding CI versus CD should be re-evaluated not merely as a data processing choice, but as a structural compromise necessitated by Non-Autoregressive (NAR) or Prefix-based architectures. Prevalent CI-based models (e.g., PatchTST) implicitly factorize the prediction target into independent distributions conditional solely on the lookback window (i.e., $P(Y|X) \approx \prod P(y_t|X)$), just mapping the history to prediction. While this simplification mitigates the noise accumulation inherent in direct mapping and offers robustness on datasets with weak inter-series correlations, it fundamentally neglects the joint probability of future trajectories and the intricate interplay of multi-channel modalities.

Consequently, we argue that for MTSF tasks exhibiting strong channel heterogeneity, the expressivity of NAR-based CI methods is theoretically bounded. Although recursive Autoregressive (AR) methods may inherently risk accumulated errors, their superior model capacity dictates that expanding the prediction window can yield lower error rates under an equivalent computational budget. Furthermore, by integrating dedicated channel interaction modules, AR architectures effectively adapt to complex inter-channel relationship modeling, overcoming the limitations of independent distribution assumptions and rendering them indispensable for strongly coupled channels. CPiRi exemplifies this paradigm shift: by anchoring a channel-interaction spatial module on a robust AR foundation (Sundial), it explicitly restores the modeling of joint spatiotemporal distributions that purely CI-based approaches fundamentally discard.

### A.5 REPRODUCIBILITY

### A.5.1 SOURCE CODE AND USAGE INSTRUCTIONS

The complete source code for CPiRi, along with all scripts required to reproduce the experiments, is released at the Github Repository JasonStraka/CPiRi. The repository is structured to facilitate ease of use:

- `/tutorial/` and `/scripts/`: Contains scripts for downloading and preprocessing all datasets according to the standardized formats used in our experiments.

- `/experiments/`: Includes the main experiment runner script (`train.py`) and configuration files for training and evaluating CPiRi and all baseline models on each dataset.

- `/baselines/`: Provides the source code for the CPiRi architecture as well as the implementations of all baseline models integrated within the framework.

- `README.md`: A detailed guide is provided with step-by-step instructions on how to set up the required software environment, prepare the data, and execute the training and evaluation scripts to reproduce the paper's results.

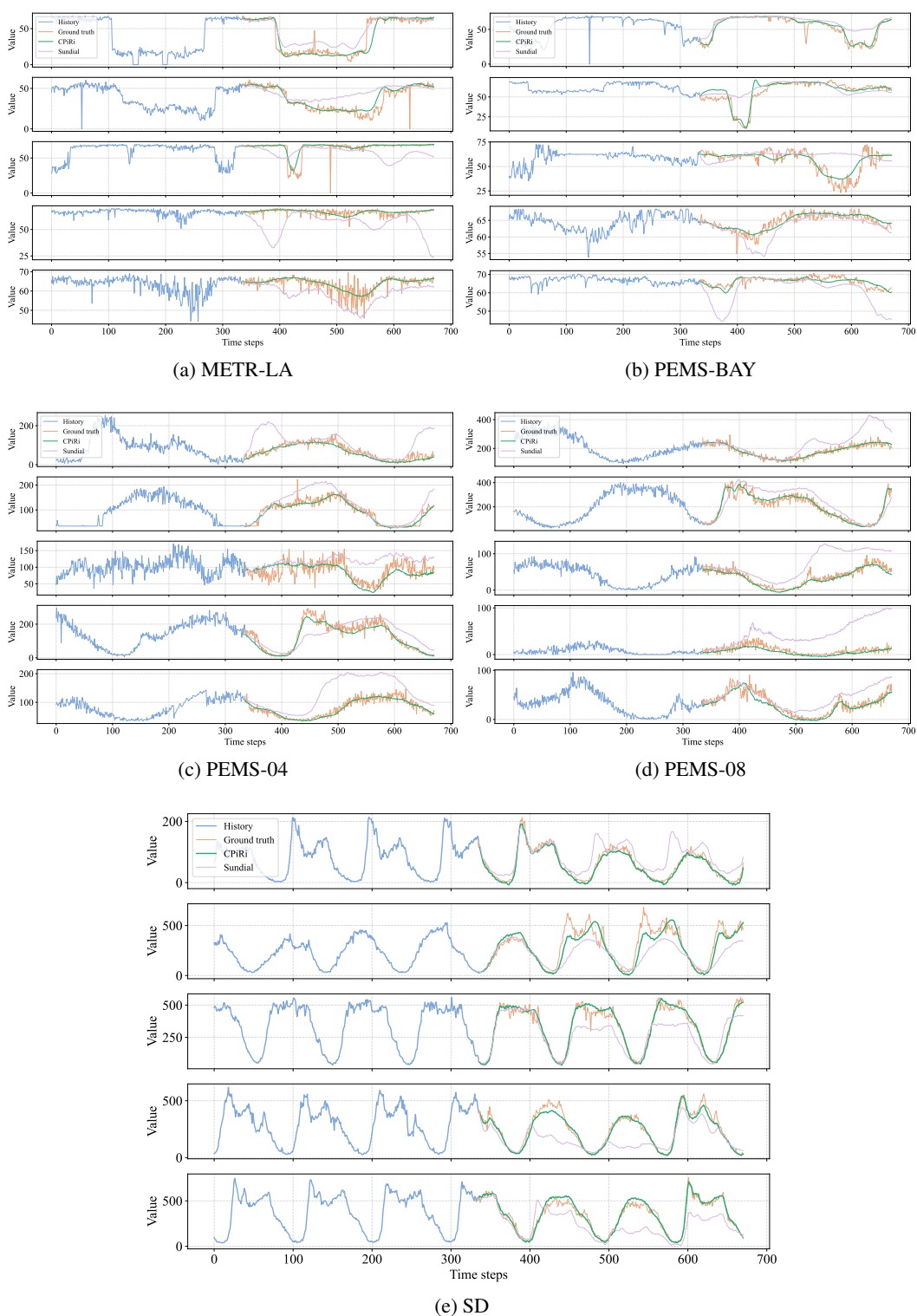

Figure 10: Forecasting case studies on five benchmark datasets. Each subplot visualizes a 336-step forecast for four selected channels, comparing CPiRi's predictions against the ground truth and the Sundial baseline. CPiRi's forecasts maintain high fidelity across various patterns, demonstrating its effective relational learning, whereas the channel-independent Sundial baseline shows clear deviations.

A.5.2   PSEUDOCODE OF THE CPiRi TRAINING PROCEDURE

Algorithm 1 in the main paper provides a high-level overview of the permutation-invariant regularization strategy. To complement this, Algorithm 2 below offers a more detailed, step-by-step description of the CPiRi model's forward pass during a single training iteration. This pseudocode serves as a formal and unambiguous specification of our method, clarifying the flow of data through the three architectural stages and the precise point at which channel shuffling is applied. This level of detail is intended to eliminate any ambiguity and facilitate replication by other researchers.

---

**Algorithm 2** CPiRi Training Procedure with Permutation-Invariant Regularization

---

**Require:** Training dataset $\mathcal{D} = \{(\mathcal{X}^{(i)}, \mathcal{Y}^{(i)})\}_{i=1}^{N}$ where $\mathcal{X} \in \mathbb{R}^{B \times L \times C}$
  Frozen temporal encoder $E_{\text{frozen}}$, trainable spatial module $M_\theta$, frozen decoder $D_{\text{frozen}}$
  Channel permutation set $\Pi_C$ (symmetric group of order $C$)
**Ensure:** Trained spatial module parameters $\theta^*$ with permutation-invariant property
 1: Initialize $\theta$
 2: **for** epoch $= 1$ **to** $K$ **do**
 3:   **for** batch $(\mathcal{X}, \mathcal{Y}) \sim \mathcal{D}$ **do**
 4:     Apply channel shuffling:
 5:       Sample random permutation $\pi \sim \text{Uniform}(\Pi_C)$
 6:       $\mathcal{X}_\pi \leftarrow \text{PermuteChannels}(\mathcal{X}, \pi)$
 7:       $\mathcal{Y}_\pi \leftarrow \text{PermuteChannels}(\mathcal{Y}, \pi)$
 8:     **Stage 1: Temporal Feature Extraction**
 9:     **for** $c = 1$ **to** $C$ **do**
10:       $\mathbf{h}_c \leftarrow E_{\text{frozen}}(\mathcal{X}_\pi[:, :, c])$
11:     **end for**
12:     $\mathcal{H} \leftarrow \text{Stack}([\mathbf{h}_1, ..., \mathbf{h}_C], \dim = 2)$
13:     **Stage 2: Spatial Interaction**
14:     $\mathcal{H}' \leftarrow M_\theta(\mathcal{H})$
15:     **Stage 3: Prediction Generation**
16:     **for** $c = 1$ **to** $C$ **do**
17:       $\hat{\mathbf{y}}_c \leftarrow D_{\text{frozen}}(\mathcal{H}'[:, :, c])$
18:     **end for**
19:     $\hat{\mathcal{Y}} \leftarrow \text{Stack}([\hat{\mathbf{y}}_1, ..., \hat{\mathbf{y}}_C], \dim = 2)$
20:     Compute loss: $\mathcal{L} \leftarrow \text{Loss}(\hat{\mathcal{Y}}, \mathcal{Y}_\pi)$
21:     Update parameters: $\theta \leftarrow \text{OptimizerStep}(\theta, \nabla_\theta \mathcal{L})$
22:   **end for**
23: **end for**
24: **return**  optimal parameters $\theta^* \leftarrow \theta$

---

