# OpenReview forum: "CPiRi: Channel Permutation-Invariant Relational Interaction for Multivariate Time Series Forecasting"
_ICLR.cc/2026/Conference — ICLR 2026 Poster_

### Official Review · Reviewer_Kq3U · 2025-10-28

**Soundness:** 2
**Presentation:** 3
**Contribution:** 3
**Rating:** 6
**Confidence:** 5

**Summary:**

This paper addresses a central paradox in multivariate time series forecasting: channel-dependent (CD) models overfit to channel order rather than learning true inter-channel relationships, while channel-independent (CI) models achieve robustness by sacrificing performance, as they neglect these critical dependencies. To resolve this trade-off, the authors propose CPiRi, a novel framework that employs a spatio-temporal decoupling architecture. CPiRi leverages a frozen, pre-trained foundation model to independently extract robust temporal features, which are then fed into a lightweight, trainable spatial module. This module is specifically trained with a permutation-invariant regularization strategy (channel shuffling) to learn content-driven, permutation-equivariant relationships. This design allows CPiRi to achieve state-of-the-art forecasting accuracy while demonstrating exceptional robustness, maintaining its performance even when channel orders are permuted during inference.

**Strengths:**

1. The proposed spatio-temporal decoupling architecture, which integrates a *frozen* time series foundation model (for temporal features) with a *lightweight, trainable* spatial module (for relational learning), is a novel and efficient design.
2. The introduction of the permutation-invariant regularization strategy (channel shuffling) is a simple yet highly effective training technique to enforce the desired invariance.
3. The experiments are pointedly designed to test the central hypothesis. The 'channel shuffling robustness analysis' (Table 2) is particularly impactful, providing a stark and convincing contrast between CPiRi's stability and the fragility of other CD models.

**Weaknesses:**

1. The paper *does* apply the shuffling strategy to baselines (Table 2, "Train Shuffle"), which is a strong point. Could the authors elaborate on *why* this strategy fails to rescue models like Informer or STID?
2. The "w/o regularization strategy" ablation (Table 4) shows only a *minor* performance drop (e.g., 9.21% vs 9.14% on METR-LA; 10.80% vs 9.43% on PEMS-08). This suggests that the permutation-equivariant architecture alone provides almost all of the robustness, and the "potent" and "critical" regularization strategy actually has a minimal impact.
3. The calculation cost of CPiRi, which adds an $O(C^2)$ spatial module, is non-trivial. However, in some cases (e.g., METR-LA, Table 1), it does not achieve SOTA performance compared to lighter models like STID.

**Questions:**

1. The paper convincingly shows that CPiRi is robust to channel shuffling, while standard CD models are not. However, it's unclear how much of CPiRi's gain comes from its permutation-equivariant *architecture* versus its "permutation-invariant regularization" *training strategy*.
2. The "channel shuffling" in Figure 4 is a key experiment. The methodology is unclear. How is "25% shuffle" defined? Does it mean 25% of the channel indices are randomly permuted *among themselves*, or 25% are swapped with other random channels? This needs a precise definition.

---

> ### Author Response · Authors · 2025-11-18
> **Response to Reviewer Kq3U**
>
> Thank you for your positive feedback on our paper. We address your problems below.
>
> ---
>
> ### Q1: Could the authors elaborate on why this strategy fails to rescue models like Informer or STID?
> Thank you for your detailed comment. These models are channel-dependent (CD) and do not possess the CPI property in their architecture. For these models, which rely on fixed positional cues (like positional encodings or 1D convolutions across the channel dimension), channel shuffling introduces noise rather than acting as a regularizer. This breaks their ability to learn static relationships, leading to the poor performance seen in Table 2 ("Train Shuffle").
>
> ---
>
> ### Q2: The "w/o regularization strategy" ablation shows only a minor performance drop... This suggests... the strategy actually has a minimal impact.
> Thank you for your insightful comment. You are correct that the permutation-equivariant architecture provides the fundamental CPI property. The regularization strategy serves to enforce this property and guarantee robust generalization, especially under data scarcity. While the performance drop is minor in the full-data setting (e.g., 9.14% vs 9.23% on METR-LA), its **very value highlights in low-data regimes**, as shown in Fig. 4 and Appendix Table 10. For instance, when training on only 25% of channels ('w/o 75%') on METR-LA, the model 'w/ strategy' achieves 9.35% WAPE, while the model 'w/o strategy' degrades to 10.27%. This shows the strategy is a critical regularizer, which is especially important in data-scarce scenarios.
>
> ---
>
> ### Q3: The calculation cost... is non-trivial... it does not achieve SOTA... compared to lighter models like STID (on METR-LA).
> Thank you for your insightful comment. Our response is as follows:
> 1. **Performance**: While STID (which uses exogenous holiday features) slightly edges out CPiRi on METR-LA, CPiRi achieves SOTA on 4 of the 5 other benchmarks (PEMS-BAY, PEMS-04, PEMS-08, and the new Electricity dataset), demonstrating superior overall performance.
> 2. **Robustness vs. accuracy**: The key tradeoff is not just accuracy but robustness. As shown in Table 2, STID's performance drops under channel shuffling (8.48% -> 18.07% on METR-LA; 10.90% -> 65.18% on PEMS-08). CPiRi maintains its performance perfectly.
> 3. **Generalizability**: STID's architecture is rigid; it cannot handle changes in the number of channels. CPiRi shows strong inductive generalization, achieving competitive performance even when trained on only 25% or 50% of channels (Fig. 4).
>
> In summary, CPiRi achieves competitive SOTA accuracy while also provides perfect robustness and generalization makes it a far more practical and powerful solution for real-world dynamic systems, a tradeoff that STID fails.
>
> ---
>
> ### Q4: How much of CPiRi's gain comes from its architecture versus its training strategy.
> Thank you for your valuable comment. Our response is as follows:
> 1. **Architecture's contribution**: The permutation-invariant interaction module we propose is a novel and critical component. As shown in Table 4, when this interaction module is removed (the "w/o spatial module" variant), performance degrades significantly (e.g., WAPE increases from 9.14% to 16.51% on METR-LA and 9.43% to 22.69% on PEMS-08 ). This demonstrates that our proposed module is both innovative and effective.
> 2. **Strategy's contribution**: The regularization strategy serves to enforce this property and guarantee robust generalization, especially **under data scarcity**. While the performance drop is minor in the full-data setting (e.g., 9.14% vs 9.23% on METR-LA), its true value is revealed in low-data regimes, as shown in Fig. 4 and Appendix Table 10. For instance, when training on only 25% of channels ('w/o 75%') on METR-LA, the model 'w/ strategy' achieves 9.35% WAPE, while the model 'w/o strategy' degrades to 10.27%. This shows the strategy is a critical regularizer, which is especially important in data-scarce scenarios.
> 3. **Further comparison**: To further validate our architecture, we also **added a new ablation**: "w/ 3 layer encoder from scratch" (**Table 4**). This replaces the large Sundial encoder with a lightweight, randomly initialized encoder.
> Although its performance is weaker than the full CPiRi (e.g., 11.17% vs 9.43% WAPE on PEMS-08), it outperforms existing "from scratch" methods, such as Informer and TimeXer. This demonstrates that our architecture still functions effectively when trained from scratch and converges more readily than the full architecture without pre-training. This further proves the value of our spatio-temporal decoupled design.
>
> ---
>
> ### Q5: How is "25% shuffle" defined?
> Thank you for your detailed comment. We are sorry for the lack of clarity. We have added a precise definition in Sec 4.2 of the revised paper: "...where 25% shuffle indicates sampling 25% channels uniformly at random and applying a random permutation within this subset, while keeping the remaining channels in their original order".

---

> > ### Comment · Reviewer_Kq3U · 2025-11-26
> >
> > Thanks for your response. Since my score is already positive, I will keep my rating.

---

> > > ### Author Response · Authors · 2025-11-28
> > >
> > > Thank you for your reply and for keeping your positive rating. We truly appreciate the time and effort you put into reviewing our paper. Your helpful comments have made our work better.

---

### Official Review · Reviewer_Vucs · 2025-10-31

**Soundness:** 2
**Presentation:** 3
**Contribution:** 2
**Rating:** 4
**Confidence:** 4

**Summary:**

CPiRi is a new framework for multivariate time series forecasting that stays accurate even when input channels are reordered or changed. It combines a frozen pre-trained temporal encoder with a lightweight spatial module trained using random channel shuffling, forcing the model to learn true content-based relationships instead of memorizing positions. This design achieves state-of-the-art accuracy, strong generalization to unseen channels, and robust performance under channel permutations.

**Strengths:**

1. Clear Problem Identification: The paper clearly diagnoses a critical flaw in existing CD models using a simple "channel shuffling" diagnostic test. This test reveals that many SOTA models rely on "positional memorization," leading to catastrophic performance collapse (e.g., >400% error increase for Informer) when channel order is changed.
2. Effective and Sound Design: The "spatio-temporal decoupling" is an elegant solution. It leverages the power of a robust, pre-trained temporal model (CI strength) while using a separate, lightweight module to explicitly learn cross-channel interactions (CD strength). The channel shuffling strategy directly enforces the desired permutation-invariant property.
3. Comprehensive Experimental Validation: The experiments are thorough and directly support the claims. CPiRi achieves SOTA accuracy , remains perfectly stable under channel shuffling , and shows strong inductive generalization to unseen channels. The ablation studies (Table 4) clearly demonstrate the necessity of each component: the spatial module, the pre-trained weights, and the shuffling strategy.

**Weaknesses:**

1. The framework's success is entirely dependent on the frozen Sundial encoder. The ablation study "w/o pretrained weights" results in "complete failure". This makes it difficult to separate the contribution of the novel CPiRi training strategy from the powerful priors of the (very large) foundation model it relies on.
2. The individual components are standard: a "standard Transformer encoder block" for the spatial module and a data augmentation technique for training. The novelty is in the clever system design and training methodology that combines these parts, rather than a fundamentally new architectural mechanism.
3. By freezing the temporal encoder, the model can handle structural co-drift (changes in channel order) but cannot adapt its temporal feature extraction to new patterns or "abrupt trend shifts". This limitation is noted by the authors and means it may struggle in scenarios where the underlying temporal dynamics of the data change over time.

**Questions:**

1. Why frozen temporal encoder? Will fine-tuning help improve performance?
2. Will applying different temporal encoders, e.g., Chronos and Moment [1], affect performance?
3. Does the strategy work for even higher-dimensional time series [2]?
4. Can the authors visualize the latent representations before and after spatial encoding across different permutations?

[1] Goswami M, Szafer K, Choudhry A, Cai Y, Li S, Dubrawski A. Moment: A family of open time-series foundation models. arXiv preprint arXiv:2402.03885. 2024 Feb 6.
[2] Ni J, Wang S, Liu Z, Shi X, Zhong X, Ye Z, Jin W. U-Cast: Learning Hierarchical Structures for High-Dimensional Time Series Forecasting. arXiv preprint arXiv:2507.15119. 2025 Jul 20.

---

> ### Author Response · Authors · 2025-11-18
> **Response to Reviewer Vucs**
>
> Thank you for your feedback and comments on our paper. We address your problems below.
>
> ---
>
> ### Q1: The framework's success is entirely dependent on the frozen Sundial encoder.
> Thank you for your thoughtful comment. Our response is as follows:
> 1. **CPiRi's contribution**: We acknowledge that part of the model's performance comes from the pre-trained model, because this is a key advantage for modeling sparse spatial relations. Nonetheless, the proposed permutation-invariant interaction is novel and essential. As shown in Table 4, when the permutation-invariant interaction module is removed ("w/o spatial module"), performance degrades significantly (e.g., WAPE increases from 9.14% to 16.51% on METR-LA). This demonstrates that our proposed module is effective. Furthermore, CPiRi possesses strong robustness under structural and distributional co-drift and in channel permutation tests, demonstrating capabilities that the foundation model alone cannot achieve.
> 2. **Comparison without Sundial**: To further validate our architecture, we also **added a new ablation**: "w/ 3 layer encoder from scratch" (**Table 4**). This replaces the large Sundial encoder with a lightweight, randomly initialized encoder. Although its performance is weaker than the full CPiRi (e.g., 11.17% vs 9.43% WAPE on PEMS-08), it outperforms existing "from scratch" methods, such as Informer and TimeXer. This demonstrates that our architecture still functions effectively when trained from scratch and converges more readily than the full architecture without pre-training. This further proves the value of our spatio-temporal decoupled design.
>
> ---
>
> ### Q2: The individual components are standard... novelty is in the clever system design...
> Thank you for your helpful comment. Our core contribution is not a new attention mechanism but **a novel framework and training methodology** (spatio-temporal decoupling + permutation-invariant regularization), which solves the CI-CD paradox through system-level design. The empirical evidence such as **SOTA accuracy**, **perfect robustness under channel shuffling** (Table 2), and **strong generalization under data scarcity** (Fig. 4), all validate that this specific combination of standard components unlocks properties that current methods do not have. Furthermore, as we articulate in the final paragraph of Section 3.5, this is **not a simple patchwork**. The self-attention mechanism is a canonical implementation of a permutation-equivariant function capable of fusing relationships between multiple representations. This makes it the ideal architectural choice to specifically address the challenges of structural and distributional co-drift and to enforce the CPI property by design. In summary, CPiRi achieves competitive SOTA accuracy while also provides perfect robustness and generalization, making it a far more practical and powerful solution for real-world dynamic systems, a tradeoff that current methods fails. Our future work will focus on hierarchical self-attention mechanism to further improve performance and efficiency of the spatial module.
>
> ---
>
> ### Q3: By freezing... [the model] cannot adapt its temporal feature extraction to new patterns or "abrupt trend shifts".
> Thank you for your helpful comment. We agree that this is a potential limitation, as noted in our "Limitation discussion". However, we posit that the primary limitation is the static fusion mechanism, not the frozen encoder. The Sundial model is pre-trained on a vast, diverse dataset, making its temporal feature extraction capability very robust. Our future work will focus on adaptive fusion (as mentioned in our conclusion). For example, this could involve dynamically fusing multi-scale representations from different depths of the Sundial encoder.

---

> ### Author Response · Authors · 2025-11-18
> **Response to Reviewer Vucs**
>
> ### Q4: Why frozen temporal encoder? Will fine-tuning help improve performance?
> Thank you for your valuable comment. Our response is as follows:
>
> 1. According to your comment, we have **conducted a new ablation study**: "w/ fine-tuning in last 10 epochs" (added to Table 4). The results show that fine-tuning offers only slight gains on some datasets but **increases memory consumption by ~5x**. This is because fine-tuning risks overfitting to the smaller MTSF datasets, thereby losing the robust features from Sundial's large-scale pre-training. Freezing the backbone prevents this and is far more efficient, as gradients are only required for the lightweight spatial module.
> 2. More importantly, we have **added a visualized figure** (**Fig. 6**) for the UMAP of the fine-tuned representations. The visualization clearly shows that the representations become more compact and less separable, suggesting the model is overfitting to the specific dataset and losing the generalizable, robust features from the foundation model, after fine-tuning.
> 3. This experiment confirms our hypothesis: freezing the encoder is crucial for preserving robust priors, preventing overfitting on smaller MTSF datasets, and maintaining the practical efficiency of our framework.
>
> ---
>
> ### Q5: Will applying different temporal encoders, e.g., Chronos and Moment, affect performance?
> Thank you for your insightful comment. Our response is as follows:
> 1. Sundial's "next-token prediction" architecture is perfectly suited for our decoupled framework, as it provides a single comprehensive feature vector per channel. The Moment model you mentioned is not based on a "next-token prediction" architecture and is not easily compatible with our method. For Chronos, while it uses "next-token prediction," it is designed for a short forecasting horizon (e.g., 64), which conflicts with our long-horizon (336) setting.
> 2. Nevertheless, we **added an experiment using the Chronos-2 encoder** ("w/ frozen Chronos-2 encoder" in Table 4). As predicted, its representations (trained for short-horizon tasks) transfer poorly to our long-horizon setting, yielding inferior performance. We will continue to follow the progress of pre-trained models in the field of time series forecasting.
>
> ---
>
> ### Q6: Does the strategy work for even higher-dimensional time series?
> Thank you for your thoughtful comment. Our response is as follows:
> 1. Regarding "higher-dimensional time series [2]", we have investigated Time-HD benchmark in [2]. We found that its Traffic-CA dataset is from the same source as our CA dataset. However, Time-HD downsamples the data to an hourly frequency and a shorter 168-step horizon on **7,491 channels**. Our work uses the original, more challenging 5-minute frequency and 336-step horizon with the full **8,600 channels**. The results on our CA dataset (Table 5) have already demonstrated our method's applicability to high-dimensional settings.
> 2. Moreover, we have also **added the widely-used Electricity dataset** (321 channels) to our benchmarks. As shown in the new results in Table 1, CPiRi achieves SOTA performance on the Electricity dataset as well. This provides strong evidence that our approach generalizes to other domains.
>
> ---
>
> ### Q7: Can the authors visualize the latent representations before and after spatial encoding across different permutations?
> Thank you for your valuable comment. According to your comment, we have **added Fig. 5** in the main paper. This figure visualizes the channel representations before and after the spatial module, and specifically shows the CPiRi outputs under three independent, random channel permutations. As shown in the figure, the three CPiRi UMAPs are nearly identical in their geometry and all show clearer cluster separation than the original Sundial embeddings. This provides strong visual proof that **our model has learned a truly content-driven, order-agnostic relational mapping**, just as intended.

---

### Official Review · Reviewer_AJUX · 2025-11-01

**Soundness:** 3
**Presentation:** 3
**Contribution:** 3
**Rating:** 6
**Confidence:** 4

**Summary:**

The paper proposes CPiRi, a channel permutation-invariant framework for multivariate time series forecasting. It combines a frozen pretrained temporal encoder with a lightweight spatial interaction module trained under random channel shuffling. This design makes the model rely on content-based relations instead of channel order, achieving strong generalization and state-of-the-art performance on multiple benchmarks.

**Strengths:**

1. Interesting motivation.
2. Clear and easy-to-follow writing.
3. Comprehensive theoretical analysis provides support for the proposed method.

**Weaknesses:**

1. Since this paper only uses Sundial as the temporal feature extractor, it lacks an explanation of why Sundial was chosen over other foundation models. Can this framework generalize to other pretrained models such as Chronos [1] or Moment [2]?
2. I appreciate that the paper uses high-dimensional datasets with channel heterogeneity. This is an interesting attempt for scalability analysis. However, could you also test the model on Time-HD [3]?
3. The main concern lies in efficiency (which the authors did not discuss in the main text). The framework invokes two large pretrained models and employs multi-head attention for high-dimensional inputs, which could lead to significant computational overhead.
4. iTransformer [4] also applies multi-head attention along the channel dimension and is permutation-equivariant. How does this work differ from iTransformer in terms of channel modeling?

[1] "Chronos: Learning the language of time series." arXiv preprint arXiv:2403.07815 (2024).
[2] "Moment: A family of open time-series foundation models." arXiv preprint arXiv:2402.03885 (2024).
[3] "U-Cast: Learning Hierarchical Structures for High-Dimensional Time Series Forecasting." arXiv preprint arXiv:2507.15119 (2025).
[4] "iTransformer: Inverted Transformers Are Effective for Time Series Forecasting." arXiv preprint arXiv:2310.06625 (2023).

**Questions:**

see weakness

---

> ### Author Response · Authors · 2025-11-18
> **Response to Reviewer AJUX**
>
> Thank you for your positive feedback on our paper. We address your problems below.
>
> ---
>
> ### Q1: Why Sundial? Can it generalize to other models like Chronos and Moment?
> Thank you for your insightful comment. Our response is as follows:
> 1. Sundial's "next-token prediction" architecture is perfectly suited for our decoupled framework, as it produces a single final patch representation that encapsulates the historical context for each channel. The Moment model you mentioned seems not based on a "next-token prediction" architecture and is not easily compatible with our method. For Chronos, while it is a "next-token prediction" model, it is designed for a short forecasting horizon (e.g., 64), which conflicts with our long-horizon (336) setting.
> 2. Nevertheless, we **added an experiment using the Chronos-2 encoder** ("w/ frozen Chronos-2 encoder" in **Table 4**). As predicted, its representations (trained for short-horizon tasks) transfer poorly to our long-horizon setting, performing worse than a 3-layer encoder trained from scratch. We will continue to follow the progress of pre-trained models in the field of time series forecasting.
>
> ---
>
> ### Q2: Could you also test the model on Time-HD?
> Thank you for your valuable comment. We have investigated Time-HD and found that its Traffic-CA dataset is from the same source as our CA dataset. However, Time-HD downsamples the data to an hourly frequency and a shorter 168-step horizon on **7,491 channels**. Our work uses the original, more challenging 5-minute frequency and 336-step horizon with the full **8,600 channels**. The results on our CA dataset (Table 5) have already demonstrated our method's applicability to high-dimensional settings.
>
> ---
>
> ### Q3: Concern about efficiency.
> Thank you for your thoughtful comment. We have included an efficiency analysis in the appendix of the original submission. According to your comment, we have made the following two changes to improve clarity.
> 1. We have **moved the detailed efficiency analysis to the main paper** (now **Table 6** in Sec 4.4).
> 2. We have **added iTransformer to this table** to provide a direct comparison of computational cost. This comparison highlights our model's advantages, as iTransformer, another CPI-capable CD model, requires over twice the GPU memory (17.69 GB vs. 8.00 GB) for comparable performance (Table 1). This demonstrates the significant efficiency gains from our spatio-temporal decoupled architecture.
>
> ---
>
> ### Q4: How does this work differ from iTransformer in terms of channel modeling?
> Thank you for your insightful comment. Our response is as follows:
> 1. **Architecture**: iTransformer performs spatio-temporal attention inside each layer, coupling the dimensions. CPiRi is radically decoupled: it runs the temporal encoder first, then a single spatial interaction module on the final tokens of $C$ channels.
> 2. **Efficiency**: The decoupling architecture makes CPiRi more efficient. iTransformer has $O((T \times C)^2)$ complexity, while CPiRi has $O(T^2 + C^2)$ complexity. As shown in Table 6, this results in CPiRi using less than half the GPU memory of iTransformer on the CA dataset (8.00 GB vs 17.69 GB) while supporting a larger batch size.
> 3. **Performance**: CPiRi's design not only is more efficient but also achieves stronger performance, as shown in Table 1 (e.g., 9.43% vs 10.70% WAPE on PEMS-08).
>
> We have **added relevant contents to Sec 4.2 and data to Table 6** to clarify this.

---

### Official Review · Reviewer_DSFJ · 2025-11-02

**Soundness:** 3
**Presentation:** 4
**Contribution:** 3
**Rating:** 6
**Confidence:** 4

**Summary:**

This paper tackles multivariate time series forecasting and points out that many models inadvertently entangle cross-channel interactions with the specific channel order seen in training. The authors introduce a three-stage framework: a frozen, pretrained univariate backbone to extract temporal features, a lightweight spatial module that views these features as an unordered set and models their relations with self-attention, and a frozen per-channel decoder. To enforce order agnosticism, training is done with random channel shuffling, discouraging any dependence on positional cues. On traffic style benchmarks, the approach proves effective, and the paper further shows it can generalize even when trained on only a subset of channels.

**Strengths:**

1. The problem framing is timely and interesting. The paper not only says “permutations matter” but builds an explicit diagnostic: train with fixed order, test with shuffled order, show catastrophic failure for several competitive models.
2. The proposed framework is reasonable and coherent, with the per channel frozen temporal encoder feeding a permutation aware spatial block, and the permutation based training strategy reinforcing the intended behavior.
3. The reported improvements indicate that the method actually delivers robustness rather than just matching accuracy in the standard setting.

**Weaknesses:**

1. My main concern is the reliance on a large pre-trained backbone. Table 4 shows that removing the pre-trained weights leads to a substantial drop in accuracy, which suggests that much of the gain comes from the foundation model rather than from the proposed permutation invariant interaction itself. However, in Table 1 the competing CD baselines are trained from scratch and do not benefit from comparable pre-training. This raises a fairness question: to what extent are the improvements due to the architectural idea, and to what extent are they due to access to stronger prior knowledge? It would help to include baselines equipped with similar pre-trained features, or to report results for CPiRi without pre-training in Table 1.
2. All benchmarks are traffic datasets. It is unclear whether the same channel ordering effect appears in other multivariate settings (e.g., energy, industrial telemetry, climate) or in higher-dimensional public datasets such as [1]. Showing at least one non-traffic dataset would clarify how general the phenomenon is.
3. The paper shows that changing channel order can hurt performance, but it is not clear how this effect scales as the number of channels grows. Do larger channel sets make models more brittle to reordering, or does the effect plateau? A controlled study where channel count is progressively increased would make the robustness claim stronger.


[1] U-Cast: Learning Hierarchical Structures for High-Dimensional Time Series Forecasting. 2025

**Questions:**

The questions are included in the weaknesses.

---

> ### Author Response · Authors · 2025-11-18
> **Response to Reviewer DSFJ**
>
> Thank you for your positive feedback on our paper. We address your problems below.
>
> ---
> ### Q1: Reliance on a large pre-trained backbone.
> Thank you for your valuable comment. Our response is as follows:
> 1. **CPiRi's contribution**: We acknowledge that part of the model's performance comes from the pre-trained model, because leveraging strong priors is a key advantage for modeling sparse spatial relations. Nonetheless, the permutation-invariant interaction module is a novel and critical component. As shown in Table 4, when this interaction module is removed (the "w/o spatial module" variant), performance degrades significantly (e.g., WAPE increases from 9.14% to 16.51% on METR-LA and 9.43% to 22.69% on PEMS-08 ). This demonstrates that our proposed module is effective.
> 2. **Fairer comparison**: To address the "from scratch" comparison, we have **added a new ablation**: "w/ 3 layer encoder from scratch" (Table 4). This replaces the large Sundial encoder with a lightweight, randomly initialized encoder. Although its performance is weaker than the full CPiRi (e.g., 4.28% vs 3.90% WAPE on PEMS-BAY), it outperforms existing "from scratch" methods, such as Informer and TimeXer. This demonstrates that our architecture still functions effectively and converges more readily than the full architecture without pre-training. This result further demonstrates the effectiveness of our spatio-temporal decoupling architecture itself.
> ---
> ### Q2: Benchmarks are all traffic datasets.
> Thank you for your insightful comment. Our response is as follows:
> 1. According to your comment, we have **added the widely-used Electricity dataset** (321 channels) as our benchmarks. As shown in the updated results in **Table 1**, CPiRi achieves SOTA performance on the Electricity dataset as well, outperforming all CI and CD baselines. This provides strong evidence that our approach is not limited to traffic data and generalizes to other domains.
> 2. Regarding "higher-dimensional public datasets such as [1]", we have investigated Time-HD benchmark in [1]. We found that its Traffic-CA dataset is the same source as our CA dataset. However, Time-HD uses a down-sampled hourly frequency and a shorter 168-step horizon on **7,491 channels**. Our work uses the original, more challenging 5-minute frequency and a 336-step forecast horizon with the full **8,600 channels**. The results on our CA dataset (**Table 5**) have already demonstrated our method's applicability to high-dimensional settings.
> ---
> ### Q3: How does the channel ordering effect scale as the number of channels grows?
> Thank you for your thoughtful comment. We apologize for not analyzing this aspect in more detail in the original revision. As seen in the channel shuffling experiments (Table 2), the datasets have varying channel counts. We reorganized the table and data in ascending order of the data channels from left to right, as shown below. We observed that models like Informer and STID show catastrophic performance degradation when shuffled on PEMS-08 (170 channels) and PEMS-04 (307 channels). However, the degradations on METR-LA (207 channels) and SD (716 channels) are comparatively less severe. This suggests that the channel count itself may not be the primary factor. We have added analysis to Sec 4.2 suggesting that the degradation pattern aligns more with the strength of inter-channel correlations and heterogeneity rather than the number of channels.
>
> |**Model**|**Shuffle**|**PEMS-08**(C=170)|**METR-LA**(C=207)|**PEMS-04**(C=307)|**PEMS-BAY**(C=325)|**SD**(C=716)|
> |:------------:|:-----------:|------------:|------------:|------------:|-------------:|--------:|
> |Informer|Test|118.19%|20.19%|83.53%|9.99%|19.55%|
> ||Train|74.00%|16.48%|49.39%|7.30%|17.38%|
> |CrossGNN|Test|17.01%|12.85%|17.95%|5.20%|19.51%|
> ||Train|16.79%|12.78%|17.92%|5.19%|19.49%|
> |TimeXer|Test|16.74%|13.79%|17.22%|5.92%|18.46%|
> ||Train|15.96%|11.84%|16.72%|4.86%|14.77%|
> |iTransformer|Test|10.70%|11.27%|12.99%|4.21%|12.45%|
> ||Train|10.58%|11.50%|13.02%|4.21%|12.40%|
> |STID|Test|65.18%|18.07%|52.31%|7.20%|12.51%|
> ||Train|11.82%|10.11%|13.75%|4.30%|12.98%|
> |Crossformer|Test|39.85%|18.06%|43.83%|6.66%|12.50%|
> ||Train|12.82%|9.87%|14.75%|4.47%|12.85%|
> |__CPiRi__(ours)|Test|10.08%|9.23%|11.93%|4.02%|13.46%|
> ||Train|9.43%|9.14%|11.67%|3.90%|12.25%|

---

### Author Response · Authors · 2025-11-18
**Response to All Reviewers**

We extend our sincere gratitude to all reviewers for their thorough analysis and constructive feedback. Your insightful comments have been valuable in strengthening our paper. We have carefully revised the manuscript to address every point raised. The changes are **highlighted**.

---

### Meta-Review · Area_Chair_j3rY · 2026-01-02

**Summary:**

The reviewers raised a series of core concerns that inform the final decision, including the extent to which the pretrained encoder affects the final performance, the performance of the proposed method on other datasets, the computational overhead of the model used in the paper, and the effectiveness of the proposed techniques.

**Reviewer Concerns:**

All key reviewer concerns were addressed by the rebuttal:
1. **The extent to which the pretrained encoder affects the final performance:** To address the "from scratch" comparison, the authors added a new ablation: "w/ 3 layer encoder from scratch" (Table 4).
2. **The performance of the proposed method on other datasets:** The authors added experiments on the widely-used Electricity dataset as well as the Time-HD benchmark.
3. **The computational overhead of the model used in the paper:** The authors moved the detailed efficiency analysis to the main paper (now Table 6 in Sec 4.4).
4. **The effectiveness of the proposed techniques:** The authors added a substantial number of additional experiments to demonstrate the effectiveness of their method.

No concerns remain outstanding.

**Reviewer Scores:**

Reviewer DSFJ, AJUX, and Kq3U scored **6**, and Reviewer Vucs scored **4**. There are no clear indications that these reviewers will revise their final scores.

---

### Decision · Program_Chairs · 2026-01-26

Accept (Poster)